# ZIKV infection induces robust Th1-like Tfh cell and long-term protective antibody responses in immunocompetent mice

Huabin Liang[1,2,5], Jinyi Tang[1,2,5], Zhihua Liu[1,2,5], Yuanhua Liu [1], Yuanyuan Huang[1,2], Yongfen Xu[1], Pei Hao [1], Zhinan Yin[3], Jin Zhong[1], Lilin Ye[4], Xia Jin[1,2] & Haikun Wang[1,2]

Induction of long-lived antibody responses during infection or vaccination is often essential for subsequent protection, but the relative contributions of T follicular helper (Tfh) cells and T helper 1 (Th1) cells for induction of antigen specific antibody responses to viruses are unclear. Here, we establish an acute Zika virus (ZIKV) infection model in immunocompetent mice, and show that ZIKV infection elicits robust Th1-like Tfh cell and protective antibody responses. While these Th1-like Tfh cells share phenotypic and transcriptomic profiles with both Tfh and Th1 cells, they also have unique surface markers and gene expression characteristics, and are dependent on T-bet for their development. Th1-like Tfh cells, but not Th1 cells, are essential for class switching of ZIKV-specific IgG2c antibodies and maintenance of long-term neutralizing antibody responses. Our study suggests that specific modulation of the Th1-like Tfh cell response during infection or vaccination may augment the induction of antiviral antibody response to ZIKV and other viruses.

[1] CAS Key Laboratory of Molecular Virology and Immunology, Institut Pasteur of Shanghai, Chinese Academy of Sciences, Shanghai, China. [2] University of Chinese Academy of Sciences, Beijing, China. [3] The First Affiliated Hospital, Biomedical Translational Research Institute, Guangdong Province Key Laboratory of Molecular Immunology and Antibody Engineering, Jinan University, Guangzhou, China. [4] Institute of Immunology, Third Military Medical University, Chongqing, China. [5]These authors contributed equally: Huabin Liang, Jinyi Tang, Zhihua Liu. Correspondence and requests for materials should be addressed to X.J. (email: xjin@ips.ac.cn) or to H.W. (email: hkwang@ips.ac.cn)

Recent global Zika virus (ZIKV) outbreaks in Brazil and more than 40 other countries revealed a marked association between ZIKV infection and increased incidence of Guillain-Barre Syndrome in adults and severe fetal development abnormalities including microcephaly[1], making the development of therapeutic drugs and preventative vaccines a global public health priority. In murine and non-human primate models, several ZIKV vaccine prototypes have been developed[2–9], and varying degrees of protection were observed with these vaccines. Other than effector T cells, antibodies are the major contributor to vaccine-induced protection. Both T follicular helper (Tfh) cells and T helper 1 (Th1) cells are thought to be essential for the elicitation of antibody responses[10], however, the relative contribution of these two subsets of T cells has recently become controversial[11].

Naive CD4+ T helper cells can differentiate into Th1 and Tfh cells during viral infection[10,12]. Th1 cells express high level of transcription factor T-bet, and produce interferon-γ (IFN-γ); they control virus spread, and provide help to the generation and maintenance of cytotoxic T lymphocytes[13]. In contrast, Tfh cells express transcription factor Bcl6[14–16], chemokine receptor CXCR5, and molecules that are critical for B cell help such as inducible T-cell co-stimulator (ICOS), CD40 ligand (CD40L), and cytokines IL-21 and IL-4[12]. Interestingly, T-bet, the key transcription factor of Th1 cells, is also expressed in Tfh cells during viral infection[17,18]. However, the function of T-bet in Tfh cell development still remains controversial. It has been reported that T-bet represses the development of Tfh cells by antagonizing the expression of multiple Tfh cell-defining genes[19]. However, a transient T-bet expression appears to be important during early Tfh cell differentiation and it enables Tfh cells to produce IFN-γ[17,18]. Therefore, the exact functions of T-bet in Tfh cells need to be further investigated.

IgG2 is the predominant protective neutralizing antibody subclass in experimental viral infections in murine models. However, whether IgG2 response is facilitated by Th1 cells or Tfh cells remains an open question. In a mixed cell culture experiment, Th1 cells were found to induce IgG2 isotype class switching when in contact with B cells[20]. Further study demonstrated that Th1 cells and B cells interact at the T-B border, leading to IgG2 isotype class switching following influenza vaccination in mice in the absence of Tfh cells[11]. These data create a conflict with the current dogma that Tfh cells are the main helper to B cells for antibody production. Although Tfh cells are known to express cytokines IL-4 and IL-21, accumulating data have also showed that Tfh cells can produce IFN-γ[17,18,21]. Whether IFN-γ-producing Tfh cells play roles in IgG2 isotype class switching is uncertain.

To investigate the relative contribution of Tfh cells and Th1 cells to the induction of antibody responses, we here characterize the Tfh- and B-cell responses longitudinally in an acute ZIKV infection model in immunocompetent mice. We find that Tfh cells are indispensable for the development of protective neutralizing antibodies against ZIKV, as well as class switching of IgG2c antibody in C57BL/6 mice (and IgG2a in BALB/c mice). Intriguingly, ZIKV infection results in a large expansion of IFN-γ-producing Th1-like Tfh cells, which bear the phenotypic and transcriptomic features of both Th1 cells and Tfh cells, but are identical to neither. Functionally, this unique cell population mediates IgG2 antibody class switching and depends on an intact IFN-γ signaling pathway.

## Results

**IFNα/β response modulates ZIKV antibody induction.** Adult immunocompetent wild-type (WT) mice are resistant to ZIKV

infection due principally to their ability to mount type I interferon responses[22], thus most recent studies on ZIKV have opted to use type I interferon deficient mice. To examine the full range of host immune responses to this virus in normal mice, we administered an anti-IFNα/β receptor antibody (anti-IFNAR1) 1 day prior to a single inoculation of ZIKV to BALB/c mice and detected a transient viremia from 2 to 3 days post infection (dpi) (Supplementary Fig. 1). In the same animals, ZIKV neutralizing antibody (nAbs) responses were detected as early as 7 dpi and lasted until 28 dpi (Fig. 1a); the nAbs plaque reduction neutralization test $(PRNT)_{50}$ titers reached an average of 1408 on 7 dpi, and further increased to 1829 and 2365 on 14 and 28 dpi, respectively; in comparison, mice not treated with the anti-IFNAR1 antibody had no appreciable nAbs on 7 dpi, and much lower $PRNT_{50}$ titers on 14 and 28 dpi (192 and 136, respectively) (Fig. 1b). We further assessed the composition of ZIKV envelope protein specific binding antibodies by ELISA, and found a progressive increase in the amount of IgM and IgG of all subclasses (IgG1, IgG2a, IgG2b, and IgG3), but not IgA; with IgG2a constituting ~2/3 of the total IgG (Fig. 1c). Consistent with having the highest level of neutralizing antibodies, adoptive transfer of sera obtained from mice immune-modulated with the anti-IFNAR1 antibody conferred the best protection against ZIKV challenge in type I/II interferon receptor-deficient ($Ifnagr^{-/-}$) C57BL/6 (AG6) mice, as gauged by weight loss (Fig. 1d) and survival rate (Fig. 1e).

**ZIKV infection induces a robust Th1-like Tfh response.** Because Tfh cell response is essential for the induction of antibody responses, we next characterized the dynamics of pre-Tfh, Tfh, and GC B-cell response in different experimental groups. We found that immune-modulated ZIKV infection with the anti-IFNAR1 antibody resulted in more robust induction of Tfh ($CXCR5^{high}PD-1^{high}$) and pre-Tfh ($CXCR5^{medium}PD-1^{medium}$) than direct ZIKV infection without antibody treatment or uninfected control. Both Tfh and pre-Tfh cells peaked on 7 dpi, accounting for 9.7% and 30.3% of the total activated CD4+ T cells, respectively, and then gradually reduced on days 14 and 28 (Fig. 2a). Similarly, GC B cell ($B220^+PNA^+FAS^+$) response in immune-modulated ZIKV infection followed the same trend as the Tfh and pre-Tfh cell responses, with a peak response of 12.7% of total B cells on 7 dpi, and then declined on days 14 and 28 (Supplementary Fig. 2), kinetically consistent with the rapid elicitation of nAb responses (Fig. 1a, b). Interestingly, we observed significantly elevated Tfh and GC B-cell responses on 14 and 28 dpi without immune-modulation (Fig. 2a; Supplementary Fig. 2), implying a delayed or less pronounced ZIKV replication below detection limit might exist (Supplementary Fig. 1b). In parallel experiments, we also included inactivated ZIKV as control. In the presence of anti-IFNAR1 antibody pre-treatment, inactivated ZIKV induced fewer Tfh cells, or cells of other helper T cell subsets, and much less antibody response, compared with live ZIKV infection (Supplementary Fig. 3). These data suggest that the robust Tfh cell and antibody responses induced by Immune-modulated ZIKV infection are dependent on ZIKV replication.

Since IFNα/β signaling pathway may be involved in Tfh development[23], we then tested whether IFNα/β receptor blockage has direct effects on Tfh cell development using the lymphocytic choriomeningitis virus (LCMV) infection model. In the absence of infection, anti-IFNAR1 treatment did not affect the differentiation of Tfh cell and GC B cell (Supplementary Fig. 4a, 4b); in the presence of LCMV infection, Tfh cell responses were enhanced, but pre-Tfh cell responses were decreased in anti-IFNAR1 treatment group compared with untreated group (Tfh cell, from 13.6 to 20.7%; pre-Tfh cell, from 31.9 to 26.4%), resulting in no significant

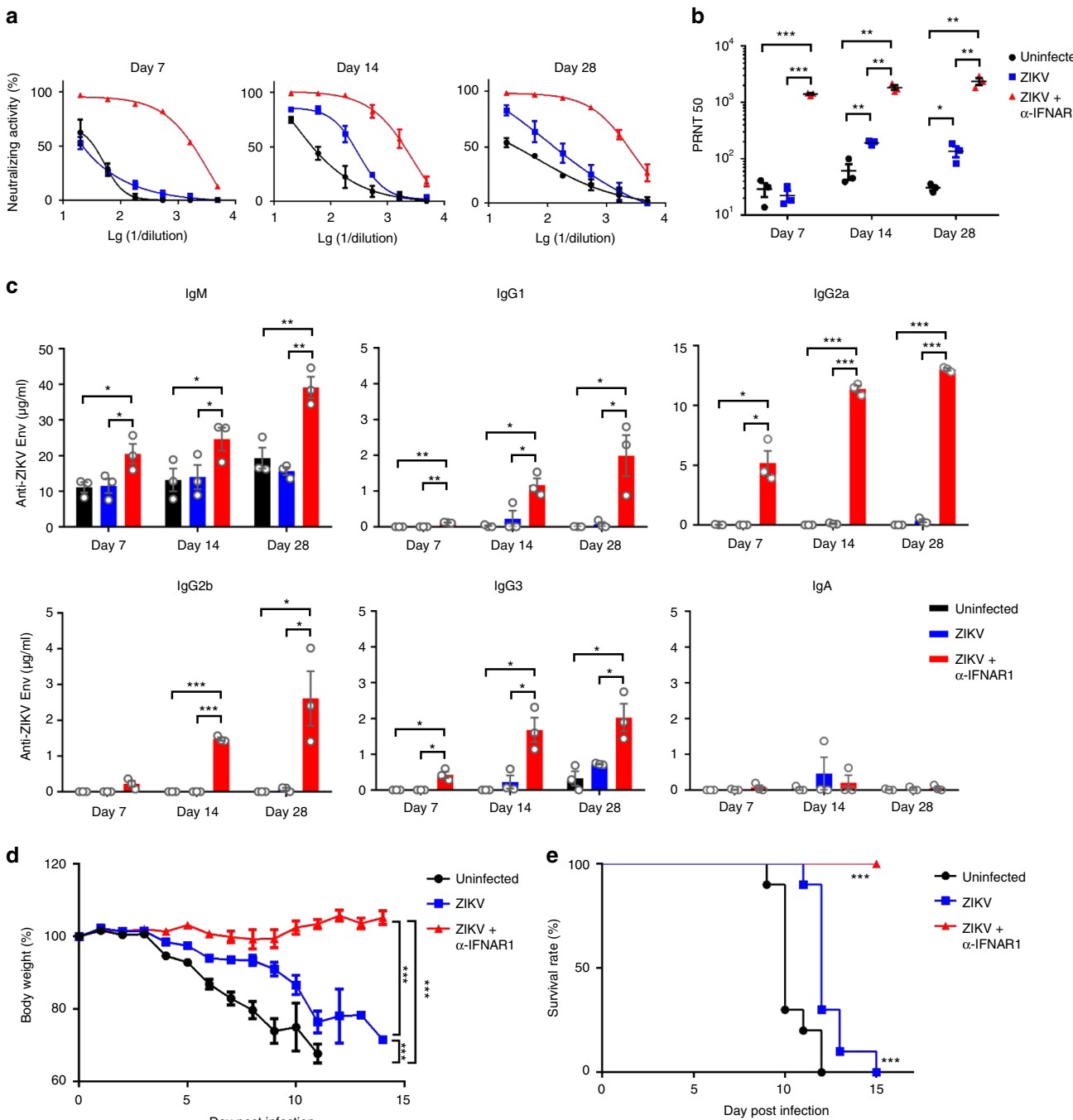

**Fig. 1** Immuno-modulation with anti-IFNα/β receptor antibody promotes the induction of protective neutralizing antibody responses after ZIKV infection in immunocompetent mice. **a–c** Three BALB/c mice per group were administered with PBS control (Uninfected), ZIKV only (ZIKV), or ZIKV + anti-IFNα/β receptor antibody (ZIKV + α-IFNAR1) ($n = 3$ for each group). **a** The kinetics of sera neutralizing antibody responses were examined on days 7, 14, and 28 post infection (dpi). **b** The PRNT$_{50}$ values in each group of mice were measured at the same time points; **c** The titers of ZIKV envelope protein specific IgM, IgG1, IgG2a, IgG2b, IgG3, and IgA antibodies in sera from each group of mice were measured by ELISA. **d**, **e** AG6 mice were adoptively transferred with 14 dpi sera (200 μl) from PBS, ZIKV, or ZIKV + α-IFNAR1 groups ($n = 10$ for each group) one day before they were challenged experimentally with ZIKV (1 × 10$^3$ PFU). The percentage of body weight (**d**) and survival rate of each group (**e**) were presented. PRNT$_{50}$ values were calculated by PROBIT analysis. The summary data were presented as mean ± standard error of mean (SEM). Statistical differences (**b**, **c**) were determined by Student's t test; Survival rate and body weight (**d**, **e**) were analyzed by log-rank test and two-way ANOVA; p values were indicated by *$p < 0.05$, or **$p < 0.01$, or ***$p < 0.001$. Data shown represent two (**a–e**) independent experiments. Source data are provided as a Source Data file

difference in total Tfh cells between the two groups (Supplementary Fig. 4c); moreover, GC B-cell responses were slightly reduced in the immune-modulated group (Supplementary Fig. 4d). Overall, the magnitude of changes in Tfh, pre-Tfh, and GC B cell responses were much less pronounced in LCMV infection than that in ZIKV infection (Fig. 2a and Supplementary Fig. 2 vs. Supplementary Fig. 4c and 4d). These data suggest that the robust Tfh, pre-Tfh, and GC B cell responses were resulted from both immune-modulation and ZIKV infection, but not caused by treatment with the anti-IFNAR1 antibody alone.

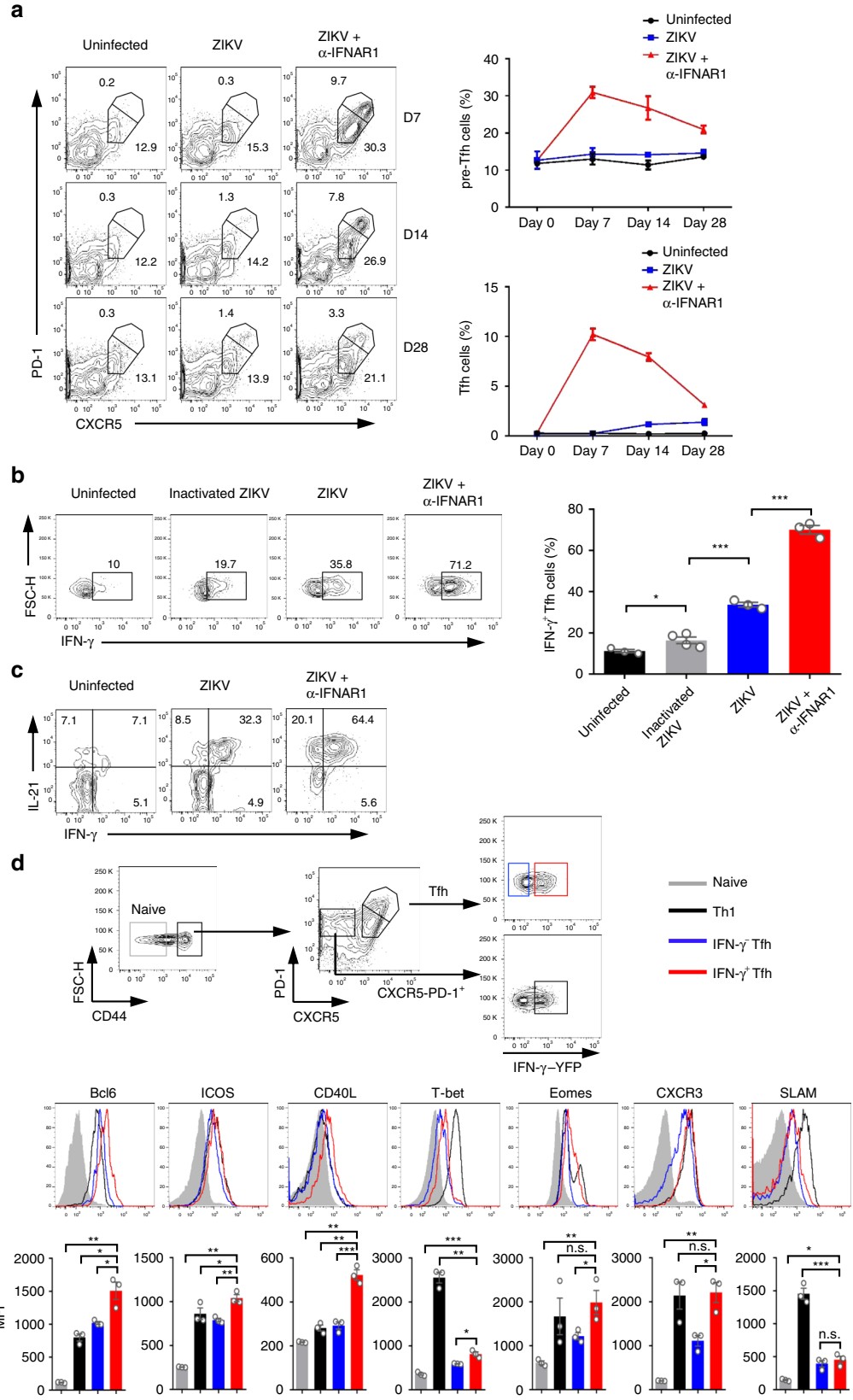

IFN-γ is known to be essential for IgG2 isotype class switching. Given the robust IgG2 antibody response observed (Fig. 1c), we next examined whether Tfh cells produce IFN-γ in ZIKV infection. Strikingly, 71.2% of Tfh cells in immune-modulated ZIKV infection expressed IFN-γ, the key cytokine of Th1 cells,

much higher than that in uninfected control (10%), those inoculated with inactivated ZIKV (19.7%), or ZIKV only (35.8%) (Fig. 2b). Since Tfh cell surface marker CXCR5 is susceptible to downregulation upon Phorbol 12-myristate 13-acetat (PMA)/ Ionomycin stimulation, to confirm that a large

**Fig. 2** ZIKV infection induces a robust Th1-like Tfh response. **a** BALB/c mice were administered with PBS control, ZIKV, or ZIKV + anti-IFNR1 as in Fig. 1, and splenocytes were harvested on 7, 14, and 28 dpi for staining with different surface markers ($n = 3$ for each group). Representative flow cytometry plots of CXCR5$^{high}$PD-1$^{high}$ Tfh cells and CXCR5$^{medium}$PD-1$^{medium}$ pre-Tfh cells gated from CD4$^+$CD44$^{high}$CD62L$^{low}$ activated T cells (left panel); with summary of the kinetic changes of Tfh cells and pre-Tfh cells (right panel). **b** BALB/c mice were administered with PBS control, inactivated ZIKV, ZIKV only, or ZIKV + anti-IFNR1, and the IFN-γ expression of CXCR5$^{high}$PD-1$^{high}$ Tfh cells were analyzed on day 7 after ZIKV infection ($n = 3-4$ for each group). Representative flow cytometry plots of IFN-γ$^+$ cells in CXCR5$^{high}$PD-1$^{high}$ Tfh cells were presented as IFN-γ$^+$ Tfh cells (left panel); and the percentages of IFN-γ$^+$ Tfh cells in each experimental group were summarized with bar charts (right panel). **c** Representative flow cytometry plot of the percentage of cells expressing IL-21 and IFN-γ in CXCR5$^{high}$PD-1$^{high}$ Tfh cells in each group of mice. **d** Gating strategy of naive cells, Th1 cells, IFN-γ$^-$ Tfh and IFN-γ$^+$ Tfh cells in IFN-γ-YFP reporter mice (top panel); the expression of Bcl6, ICOS, CD40L, T-bet, Eomes, CXCR3, and SLAM in each T cell subsets (middle panel). Mean fluorescence intensity (MFI) of each protein was summarized with bar graph (bottom panel) ($n = 3$ for each group). The summary data were presented as mean ± SEM. Statistical differences were determined by Student's t test and p values were indicated by *$p < 0.05$, or **$p < 0.01$, or ***$p < 0.001$. Data shown represent two (**a–d**) independent experiments. Source data are provided as a Source Data file

proportion of Tfh cells are indeed capable of making IFN-γ, we took advantage of CXCR5-GFP reporter mice, in which GFP protein is the indicator of CXCR5 expression[24]. In sorted Tfh cells (CD4$^+$CD44$^{high}$CD62L$^{low}$GFP$^{high}$PD-1$^{high}$), 74% of cells produced IFN-γ in immune-modulated ZIKV infection; whereas only 24% of them in ZIKV envelope protein immunization group did so (Supplementary Fig. 5a). Taken together, our results suggest that IFN-γ-producing Tfh cells were preferentially induced by replicable ZIKV infection, but not by protein, inactivated ZIKV, or non-replicable ZIKV infection.

To confirm that the observed high frequency of IFN-γ-producing Tfh cells was not caused directly by anti-IFNAR1 treatment, we repeated the LCMV infection experiment and found that the percentages of IFN-γ$^+$ Tfh and pre-Tfh cells showed no statistically significant differences between anti-IFNAR1 treated and untreated groups after LCMV infection (Supplementary Fig. 5b), thus excluding a direct role for anti-IFNAR1 treatment in the induction of IFN-γ-producing Tfh cells, and confirming that the robust IFN-γ$^+$ Tfh cell response (more than 70%) is unique to ZIKV infection, because only 33−35.7% Tfh cells produced IFN-γ in direct LCMV infection or immune-modulated LCMV infection (Supplementary Fig. 5b).

We further defined the characteristics of IFN-γ$^+$ Tfh cells. Consistent with previous reports, almost all IFN-γ$^+$ Tfh cells expressed IL-21, a pivotal functional cytokine produced in Tfh cells (Fig. 2c), implying they are functional. Next, we investigated whether IFN-γ$^+$ Tfh cells have specific phenotypic and transcriptomic profiles by comparing the expression levels of their cell-surface markers and transcription factors with those in naive T cells, Th1 cells, and IFN-γ$^-$ conventional Tfh cells (Fig. 2d), and performed similar comparisons for IFN-γ$^+$ pre-Tfh cells (Supplementary Fig. 5c). The expression levels of Bcl6, ICOS, and CD40L in IFN-γ$^+$ Tfh cells were significantly higher than those in IFN-γ$^-$ conventional Tfh cells (Fig. 2d), suggesting they may represent more activated Tfh cells and have stronger ability in B cell help compared with conventional Tfh cells. As expected, IFN-γ$^+$ Tfh cells expressed higher levels of T-bet, Eomes and CXCR3 than IFN-γ$^-$ conventional Tfh cells, but lower than Th1 cells (Fig. 2d); and T-bet was slightly downregulated in IFN-γ$^+$ Tfh cells on day 14 compared with that on day 7 after ZIKV infection (Supplementary Fig. 5d). These data suggest that IFN-γ$^+$ Tfh cells are not Th1 cells despite manifested Th1 phenotypic characteristics. Thus, we herein refer the IFN-γ$^+$ Tfh as Th1-like Tfh cells.

**Th1-like Tfh cells have features of both Tfh and Th1 cells.** To investigate whether Th1-like Tfh cells are a unique helper T cell subset, we analyzed the global gene expression of Th1-like Tfh cells in comparison with conventional Tfh cells and Th1 cells, and identified 1245 genes differentially expressed between Tfh cells, Th1 cells and Th1-like Tfh cells. Th1-like Tfh cells manifested distinctive gene expression profiles

compared with Tfh and Th1 cells, despite sharing some similar gene clusters with them (Fig. 3a). Most Tfh cell-specific genes (such as *Bcl6*, *Cxcr5*, *IL21*, and *Ascl2*) and Tfh cell-associated genes (such as *Tcf7*, and *Matk*) had a similar expression level between Tfh cells and Th1-like Tfh cells (Fig. 3b, c). However, many Th1 cell-specific genes (including *Tbx21*, *Cxcr3*, and *Ifng*) had also been upregulated in Th1-like Tfh cells (Fig. 3b, d), indicating Th1-like Tfh cells have phenotypes of both Tfh cells and Th1 cells. Specifically, 64 genes were upregulated, including Th1-associated gene *Crtam*; 83 genes were down-regulated, including Tfh-associated gene *IL4* (Fig. 3e), suggesting Th1-like Tfh cells are functionally distinct from conventional Tfh cells.

For pathway enrichment analysis of upregulated genes in Th1-like Tfh cells compared with conventional Tfh cells, we found significant enrichment of 23 pathways, including some that are important for Th1 cell, 'IL-12 and Stat4 dependent signaling pathway in Th1 development' and others critical for immune responses, such as 'cellular response to interferon beta', 'T cell migration', 'T cell costimulation', and 'leukocyte transendothelial migration' (Fig. 3f). Interestingly, some enriched pathways are linked to virus control, such as 'negative regulation of viral genome replication'. Some others are more generally related to cell cycles, for example, 'DNA dependent DNA replication', 'sister chromatid segregation', 'DNA replication initiation', and 'cell cycle checkpoint', suggesting that Th1-like Tfh cells are a more proliferative population compared with conventional Tfh cells (Fig. 3f, g).

Of note, Fang et al. recently showed a transient T-bet expression in IFN-γ$^+$ Tfh cells upon peptide immunization, and these cells are referred to as "ex-T-bet" Tfh cells[17]. To examine whether Th1-like Tfh cells and "ex-T-bet" Tfh cells are in fact the same population of cells, we compared our RNA-seq data set of Th1-like Tfh cells with that of "ex-T-bet" Tfh cells in the study of Fang et al.[17]. Comparative analysis revealed that Th1-like Tfh cells were transcriptomically distinct from the "ex-T-bet" Tfh cells (Supplementary Fig. 6a, 6b), for example, *Klrk1* gene encoding Natural Killer Group 2D (NKG2D) was significantly lower in Th1-like Tfh cells (Supplementary Fig. 6b). Consistently, Th1-like Tfh cells were also phenotypically distinct from the "ex-T-bet" Tfh cells, for example, Th1-like Tfh cells expressed high levels of the Th1 cell markers T-bet and CXCR3 (Supplementary Fig. 5d; Supplementary Fig. 6c); but low levels of the signature marker of "ex-T-bet" Tfh cell, NKG2D, compared with "ex-T-bet" Tfh cells published by Fang et al. (Supplementary Fig. 6d)[17]. These data suggest that Th1-like Tfh cells observed in ZIKV infection are not the "ex-T-bet" Tfh cells detected in peptide immunization.

**Th1-like Tfh cells help neutralizing antibody induction.** In addition to Tfh cells, Th1 cells have been reported to provide cognate B cell help, especially for the generation of IgG2

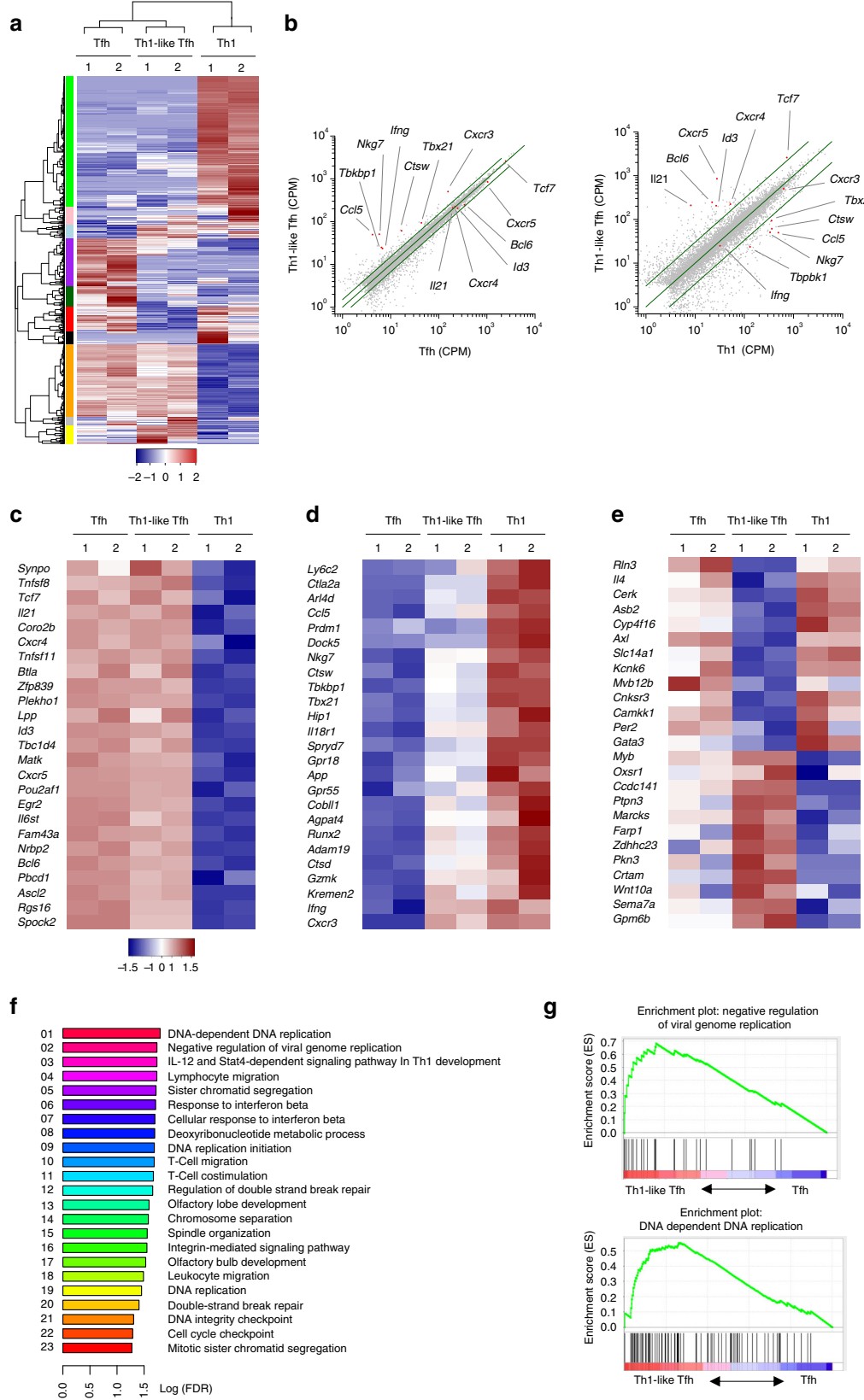

antibodies[11]. In ZIKV infection, we also observed an elevated IFN-γ-producing Th1 cell response, from 5.7% at baseline to 9.8% in ZIKV infection, and 14.1% in immune-modulated ZIKV infection, whereas Th2 or Th17 cell responses were not augmented (Supplementary Fig. 7). To address whether Tfh cells or Th1 cells are the dominant T cell subset in providing help to the elicitation of ZIKV-specific neutralizing antibody responses, we infected *Bcl6*fl/fl mice (WT) and *Bcl6*fl/fl*Cd4*-Cre mice (that

**Fig. 3** Th1-like Tfh cells have characteristics of both Tfh and Th1 cells. IFN-γ-YFP reporter mice were administered with ZIKV with or without anti-IFNAR1 blocking antibody as described in Fig.1. Splenocytes were harvested on 7 dpi for RNA-seq analyses. **a** Heatmap of the significantly different expression of global genes (rows) in IFN-γ⁻ Tfh (Tfh; replicates 1 and 2), Th1-like Tfh (Th1-like Tfh; replicates 1 and 2), and Th1 (Th1; replicates 1 and 2) cells (columns). BH-adjusted $p$ value (FDR), $\alpha < 0.05$, 1.5-fold change in Th1-like Tfh cells vs. Tfh cells, threefold change in Th1-like Tfh cells vs. Th1. **b** Scatter plot analysis of the expression of Tfh cell (left) and Th1 cell (right)-related genes in Th1-like Tfh cells compared with IFN-γ⁻ (adjusted $p$-value < 0.05, fold change > 1.5) and Th1 cells (adjusted $p$-value < 0.05, fold change > 3). **c, d** Heatmap of the significantly different expression of **c** Tfh-related genes (rows), **d** Th1-related genes (rows) in IFN-γ⁻ Tfh (Tfh; replicates 1 and 2), Th1-like Tfh (Th1-like Tfh; replicates 1 and 2), and Th1 (Th1; replicates 1 and 2) cells (columns). FDR, $\alpha < 0.05$. **e** Heatmap of the significantly different expression of genes (rows) in Th1-like Tfh cells (Th1-like Tfh; replicates 1 and 2) compared with IFN-γ⁻ Tfh (Tfh; replicates 1 and 2) and Th1 (Th1; replicates 1 and 2) cells (columns). FDR, $\alpha < 0.05$. **f** Enrichment of upregulated pathways in Th1-like Tfh cells, with the top 23 pathways listed. FDR $q$-value < 0.05. **g** Enrichment of gene signatures related to the Kyoto Encyclopedia of Genes and Genomes Negative Regulation of Viral Genome Replication (up, NES is 2.026; FDR $q$-value is 0.0183) and DNA Dependent DNA replication pathway (down, NES is 2.112; FDR $q$-value is 0.0159) in Th1-like Tfh cells. Data (**a–g**) are pooled from two independent technical duplicates

have Tfh cell developmental defect) with ZIKV after anti-IFNAR1 antibody pre-treatment. Consistent with previous findings, no obvious defects were observed in Th1 cell responses in *Bcl6*$^{fl/fl}$*Cd4*-Cre mice (Fig. 4a), but Tfh and pre-Tfh cells and GC B cells could not be generated in these mice (Fig. 4b and Supplementary Fig. 8a). Concomitantly, the percentages and absolute number of IgG1, IgG2b, and IgG2c expressing B cells in IgD$^{low}$ B cells were significantly reduced, with a compensatory increase in the percentage and number of IgM$^+$ B cells (Fig. 4c). These results suggest that antibodies produced by B cells could not go through class-switch-recombination (CSR) in the absence of Tfh cells in the ZIKV infection model, and thus most B cells were arrested at the IgM producing stage rather than transitioned to an IgG producing state. Consistent with the B cell results, the total secreted ZIKV envelope specific IgG in sera was significantly reduced in the absence of Tfh cells (Fig. 4d). Furthermore, without Tfh cells, the concentration of each IgG isotype antibody decreased, including IgG1, IgG2b, IgG2c, and IgG3 (Fig. 4d). Consequently, the sera PRNT$_{50}$ titers of immune sera decreased about three-fold in *Bcl6*$^{fl/fl}$*Cd4*-Cre mice compared with WT mice on 14 dpi (Fig. 4e). Over time, the level of nAbs only slightly decreased in WT mice, from an average PRNT$_{50}$ value of 2804 on 14 dpi to 2737 on 56 dpi, and 1477 on 130 dpi; in contrast, its levels decreased more dramatically in Bcl6-deficient (*Bcl6*$^{fl/fl}$*Cd4*-Cre) mice, from an average PRNT$_{50}$ value of 1081 on 14 dpi to 357 on 56 dpi, and 164 on 130 dpi (Fig. 4e). These data suggest that Tfh cells are essential for the generation and maintenance of long-term memory of ZIKV-specific neutralizing antibody response. Importantly, immune sera collected on 130 dpi from Bcl6 intact mice protected 6 of 9 (66.7%) AG6 mice from lethal ZIKV infection; in contrast, sera from *Bcl6*$^{fl/fl}$*Cd4*-Cre mice or uninfected mice (PBS control) collected at the same time point failed to protect AG6 mice from weight loss (Fig. 4f), or death (Fig. 4g) after lethal infection. The adoptive transfer of sera collected on 14 dpi also demonstrated that immune sera from Bcl6 intact mice had better protection against weight loss and mortality caused by ZIKV infection (Supplementary Fig. 8b, 8c).

**T-bet controls Th1-like Tfh and IgG2c class switching.** Both Th1 cells and Tfh cells could make IFN-γ that is essential for IgG2 antibody production, however, which one of them facilitates IgG2 class switching remains controversial[11]. We have showed above that the ZIKV envelope specific IgG2 antibodies in sera were dramatically reduced in the *Bcl6*$^{fl/fl}$*Cd4*-Cre mice, in which the Th1 cell differentiation was normal, but Tfh and pre-Tfh cells were absent (Fig. 4a, b). These results suggest that Tfh or pre-Tfh cells, but not Th1 cells, provided the principal help for the generation of IgG2 antibody responses in ZIKV infection. Given the majority of Tfh cells were Th1-like Tfh cells in ZIKV infection,

the reduction of IgG2 antibodies are most likely due to the lack of IFN-γ$^+$ Th1-like Tfh cells.

To further demonstrate that Th1-like Tfh cells and Th1-like pre-Tfh cells are the main contributors to IgG2 class switching, we used T-bet-deficient mice in addition to *Bcl6*$^{fl/fl}$*Cd4*-Cre mice, and confirmed that Th1-like Tfh differentiation in ZIKV infection was T-bet dependent, in agreement with a recent study demonstrating that T-bet is important for IFN-γ expression in Tfh cell[18]. In T-bet-deficient mice, the percentage and total number of Tfh cells did not change significantly, with an ~45% reduction of pre-Tfh cells in ZIKV infection, however, Th1-like Tfh cells and Th1-like pre-Tfh cells almost disappeared (Fig. 5a, b). Considering T-bet-deficient mice had Tfh cells, but lacked IFN-γ$^+$ Th1-like Tfh cells, we used this unique tool to address the roles of Th1-like Tfh cells in facilitating IgG2 class switching. We found that IgG2c positive B cells and IgG2c level in sera were dramatically decreased in T-bet-deficient mice, whereas IgG1 and IgG2b producing B cells and antibody levels in sera were sharply increased, but ZIKV-specific IgA and IgM were not affected by the loss of T-bet (Fig. 5c, d). Next, we further addressed whether these functions were T cell-intrinsic by using the *Tbx21*$^{fl/fl}$*Cd4*-Cre mice. After immune-modulated ZIKV infection, the percentage of Tfh did not change, with a slight reduction of total number for Tfh cells; whereas the percentage and total number of pre-Tfh cells had a modest reduction (Supplementary Fig. 9a), which were consistent with those in *Tbx21*$^{-/-}$ mice. Consistently, fewer IgG2c-producing B cells and reduced IgG2c antibody levels were observed in these mice compared with the WT mice, whereas IgG1 production was increased. Unexpectedly, the level of IgG2b antibodies and the number of IgG2b producing B cells were decreased in *Tbx21*$^{fl/fl}$*Cd4*-Cre mice (Supplementary Fig. 9b). Collectively, our data suggested that Th1-like Tfh and Th1-like pre-Tfh cells, but not Th1 cells, are essential for IgG2c isotype class switching.

**Th1-like Tfh cells assist IgG2c antibody class switching.** To further verify the specific role of Th1-like Tfh cells in mediating IgG2c class switching, we made *Tbx21*$^{-/-}$: *Bcl6*$^{fl/fl}$*Cd4*-Cre mixed bone marrow chimeric mice. In these chimeric mice, Th1 cells and Tfh cells were normally generated after ZIKV infection, compared with *WT*: *Bcl6*$^{fl/fl}$*Cd4*-Cre mixed bone marrow chimeras (Fig. 6a, b). Given Bcl6-deficient CD4$^+$ T cells cannot differentiate into Tfh cells, all Tfh cells in *Tbx21*$^{-/-}$: *Bcl6*$^{fl/fl}$*Cd4*-Cre mixed bone marrow chimeras were from *Tbx21*$^{-/-}$ CD4$^+$ T cells that could not differentiate into Th1-like Tfh cells (Fig. 6c). Thus, by these chimeras, we could more specifically examine the effect of Th1-like Tfh cells on IgG2c isotype class switching. After ZIKV infection, *Tbx21*$^{-/-}$: *Bcl6*$^{fl/fl}$*Cd4*-Cre mixed bone marrow chimeras showed significantly fewer IgG2c-producing cells and lower levels of ZIKV envelope specific IgG2c in sera, compared

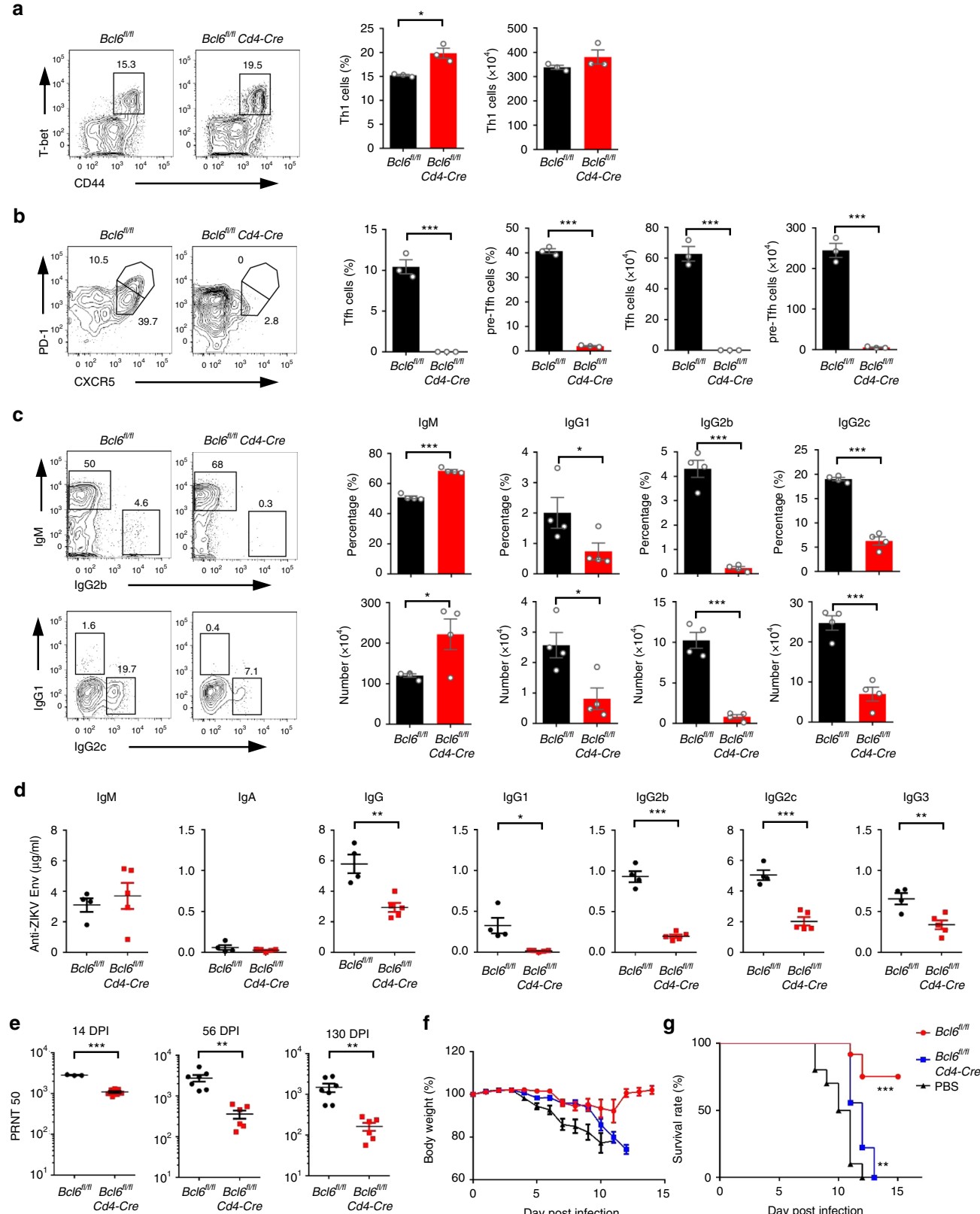

with *WT*: *Bcl6^fl/fl Cd4*-Cre mixed bone marrow chimeras (Fig. 6d, e). Consistent with observation in *Tbx21^−/−* mice, IgG1 producing cells and the level of ZIKV envelope specific IgG1 in sera markedly increased in *Tbx21^−/−*: *Bcl6^fl/fl Cd4*-Cre mixed bone marrow chimeras (Fig. 6d, e). Our data suggested that Th1-like Tfh and Th1-

like pre-Tfh cells have intrinsic function in mediating IgG2c isotype class switching.

**IFN-γ pathway is required for IgG2c class switching.** Having demonstrated that Th1-like Tfh cells are unique cells that play a

**Fig. 4** Tfh responses are indispensable for greater and long-term neutralizing antibody responses. **a-d** $Bcl6^{fl/fl}$ mice and $Bcl6^{fl/fl}Cd4$-Cre mice were administered with anti-IFNAR1 antibody one day prior to ZIKV infection ($n = 3$–5 for each group). **a** Representative flow cytometry plots of CD44$^{high}$T-bet$^+$ cells in CD4$^+$CXCR5$^-$ cells were presented as Th1 cells (left panel); bar graphs summarized the percentages and numbers of Th1 cells in each group on 7 dpi (right panel). **b** Representative flow cytometry plots of CXCR5$^{high}$PD-1$^{high}$ cells and CXCR5$^{medium}$PD1$^{medium}$ in CD4$^+$CD44$^{high}$CD62L$^{low}$ cells were presented as Tfh cells and pre-Tfh cells (left panel); bar graphs summarized the percentages and numbers of cells in each group (right panel) on 7 dpi. **c** Representative flow cytometry plots of IgM, IgG1, IgG2b, and IgG2c staining of IgD$^{low}$ B cells (left panel); bar graphs summarized the percentages and numbers of cells on 7 dpi (right panel). **d** The concentration of ZIKV envelope specific IgM, IgA, total IgG, IgG1, IgG2b, IgG2c, and IgG3 antibodies in sera from each group were tested by ELISA. **e** PRNT$_{50}$ titers of sera were measured on 14, 56, and 130 dpi in control mice group ($n = 3$ for 14 dpi, and $n = 7$ for 56 and 130 dpi) or $Bcl6^{fl/fl}Cd4$-Cre mice group ($n = 7$ for 14 dpi, and $n = 6$ for 56 and 130 dpi). **f, g** AG6 mice were adoptively transferred with sera obtained on 130 dpi (200 μl) each of the experimental groups ($n = 9$–10 for each group) at one day prior to ZIKV challenging ($1 \times 10^3$ PFU). The percentage of body weight (**f**) and survival rate (**g**) of each group were summarized. The summary data were presented as mean ± SEM. PRNT$_{50}$ values were calculated by PROBIT analysis. Statistical differences (**a-e**) were determined by Student's $t$ test; Survival rate and body weight (**f, g**) were analyzed by log rank test and two-way ANOVA; $p$ values were indicated by *$p < 0.05$, or **$p < 0.01$, or ***$p < 0.001$. Data shown represent two (**d-g**) or at least three (**a-c**) independent experiments. Source data are provided as a Source Data file

pivotal role in class switching of antibody and helping GC B cell development, to further address whether the IFN-γ pathway is essential for IgG2 antibody class switching, we used the IFN-γ receptor-deficient mice ($Ifngr1^{-/-}$) that were pre-treated with anti-IFNAR1 antibody and then inoculated with ZIKV. On 7 dpi, the numbers and percentages of Tfh cells and Th1-like Tfh cells did not change significantly in $Ifngr1^{-/-}$ mice, compared with WT mice (Fig. 7a, and Supplementary Fig. 10a); likewise, other CD4$^+$ T cell subsets, including Th1, Th2, and Th17 were also not impacted, as examined by the cytokine production profiles in $Ifngr1^{-/-}$ mice (Supplementary Fig. 10b). These results suggest that the dominant CD4$^+$ helper T cell subsets were not altered after ZIKV infection in the absence of IFN-γ signaling pathway. However, GC B cells were significantly decreased when IFN-γ pathway was deficient (Fig. 7b), suggesting that IFN-γ pathway is required for the development and formation of GC B cells.

Consistent with results in T-bet-deficient mice, $Ifngr1^{-/-}$ mice had much lower levels of IgG2b, IgG2c and IgG3 concentration in sera, and fewer IgG2b and IgG2c-producing B cells, but increased IgG1 level in sera and IgG1 producing B cells, and maintained steady levels of IgM and IgA concentration and antibody-producing B cells (Fig. 7c, d). These results indicated that IFN-γ pathway is important for IgG2 antibody class switching.

**IFN-γ requirement for class switching is B cell-intrinsic**. To further confirm that B cells require IFN-γ signaling for IgG2c class switching, we took advantage of CD45.1$^+$WT: $Ifngr1^{-/-}$ mixed bone marrow chimeras in which the ratio of WT CD4$^+$ T cells to $Ifngr1^{-/-}$ CD4$^+$ T cells, or WT B cells to $Ifngr1^{-/-}$ B cells was approximately 1:1 (Supplementary Fig. 10c). After ZIKV infection, IFN-γ receptor-deficient Tfh cells developed normally, with slight reduction of pre-Tfh cells (Fig. 8a), and these cells could produce similar levels of IFN-γ, compared with CD45.1$^+$ wild-type Tfh cells in the same chimeras (Fig. 8b), consistent with that in $Ifngr1^{-/-}$ mice (Supplementary Fig. 10a). These results suggest that IFN-γ signaling does not play significant roles in the development of Tfh and Th1-like Tfh cells in ZIKV infection. However, IFN-γ receptor-deficient B cells had much reduced IgG2c but increased IgG1 antibody responses than CD45.1$^+$ wild-type B cells in the chimeras (Fig. 8c, d), consistent with that in $Ifngr1^{-/-}$ mice (Fig. 7c). Thus, our data suggested that IFN-γ receptors on B cells are intrinsically required for IgG2c class switching.

In addition to Th1-like Tfh cells, many other cell types such as NK cells, Th1 cells, and CD8$^+$ T cells can produce IFN-γ. It raised the question of whether IFN-γ from Th1-like Tfh cells intrinsically facilitates IgG2 switching. To address the question, we made an $Ifng^{-/-}$: $Bcl6^{fl/fl}Cd4$-Cre mixed bone marrow chimeric mice, in which Tfh cells all came from $Ifng^{-/-}$ CD4$^+$

T cells. As expected, $Ifng^{-/-}$: $Bcl6^{fl/fl}Cd4$-Cre mixed bone marrow chimeras could generate Tfh cells, pre-Tfh cells, and Th1 cells normally (Fig. 8e, f), but could not generate IFN-γ$^+$ Th1-like Tfh cells, compared with WT: $Bcl6^{fl/fl}Cd4$-Cre mixed bone marrow chimeras (Fig. 8g). With the loss of IFN-γ$^+$ Th1-like Tfh cells, $Ifng^{-/-}$: $Bcl6^{fl/fl}Cd4$-Cre mixed bone marrow chimeras had fewer IgG2c-producing cells and lower level of ZIKV envelope specific IgG2c in sera after ZIKV infection (Fig. 8h, i), suggesting IFN-γ produced by Th1-like Tfh cells is responsible for IgG2 switching.

## Discussion

Viral infections generally induce both Th1 and Tfh cell responses. Although both of them provide help for antibody production, the relative contribution of these two subsets to the generation of a protective antibody response is unclear. Here, we have shown that an acute ZIKV infection in mice induces a robust response of Th1-like Tfh cells that have many Th1 characteristics, such as expressing the Th1 key transcription factor T-bet and making Th1 key cytokine IFN-γ. We also demonstrated that Th1-like Tfh cells are the main contributor to the induction and long-term maintenance of protective neutralizing antibodies against ZIKV, and that these cells are responsible for isotype class switching of the IgG2a or IgG2c antibodies that constitute the majority of ZIKV-specific antibodies.

Despite some previous experimental studies indicating the existence of IFN-γ-producing Th1-like Tfh cells[18,21], such a large expansion of Th1-like Tfh cells (more than 70% of the total Tfh cells) has never been observed before. It also appears that such a phenomenon is unique to ZIKV infection, because in a similar experimental infection using LCMV (Supplementary Fig. 5b), Th1-like Tfh cells only accounted for ~30% of the total Tfh cells (Supplementary Fig. 5b). The genesis of Th1-like Tfh cells is of some biological interests. It is known that viral infections induce both Th1 and Tfh cells under the transcriptional control of T-bet and Bcl6, respectively; however, the particular cytokine milieu may also shape their differentiation pathways. Our RNA-seq analysis revealed that many genes involved in IL-12-stat4, and IFN-γ pathways were upregulated in Th1-like Tfh cells (Fig. 3), suggesting that CD4$^+$ T cells may receive more signals from IL-12 or IFN-γ pathways provided by dendritic cells (DC) during Tfh cell differentiation in ZIKV infection.

Induction of great neutralizing antibody responses is pivotal for most vaccines. It is generally agreed that Tfh cells are the specialized helper T cells for the formation of B cell germinal centers and elicitation of long-term antibody response. However, a recent study has reported that an inactivated influenza vaccine could induce protective nAbs without Tfh and GC B cells[11]. In the ZIKV infection model, however, we found an indispensable role of Tfh cells, much more important than Th1 cells, for the

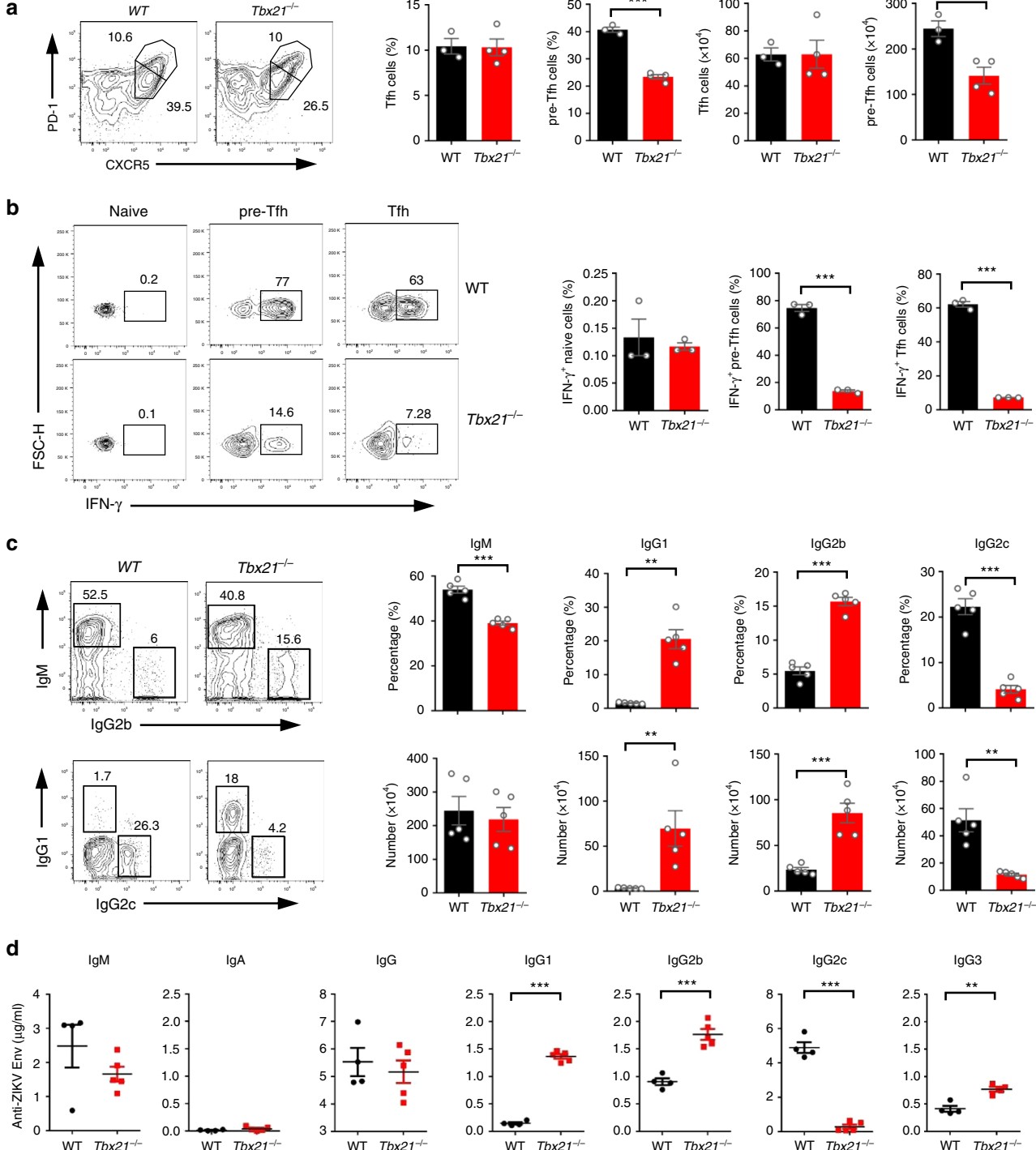

**Fig. 5** T-bet is essential for Th1-like Tfh cell differentiation and IgG2c antibody class switching. **a–d** Wild-type (WT) mice and *Tbx21−/−* mice were infected by ZIKV with anti-IFNAR1 blocking antibody. Splenocytes were collected for measurement of Tfh cell on 7 dpi and antibody-producing B-cell responses on 14 dpi, and sera were collected for antibody responses on 14 dpi (*n* = 3–5 for each group). **a** Representative flow cytometry plots of CXCR5highPD-1high cells and CXCR5mediumPD-1medium in CD4+CD44highCD62Llow T cells were presented as Tfh cells and pre-Tfh cells (left panel); bar graphs summarized the percentages and numbers of cells in each group. **b** Representative flow cytometry plots of IFN-γ staining in CD4+CD44low naive cells, CXCR5medium PD-1medium pre-Tfh cells and CXCR5highPD-1high Tfh cells (left panel); the percentages of IFN-γ+ cells in each group were summarized in bar graphs (right panel). **c** Representative flow cytometry plots of IgM, IgG1, IgG2b, and IgG2c staining of IgDlow B cells (left panel); bar graphs summarized the percentages and numbers of cells in each group. **d** The concentrations of ZIKV envelope specific IgM, IgA, total IgG, IgG1, IgG2b, IgG2c, and IgG3 antibodies were measured by ELISA. The summary data were presented as mean ± SEM. Statistical differences were determined by Student's *t* test and *p* values were indicated by **p* < 0.01, or ***p* < 0.001. Data shown represent two (**d**) or at least three (**a**–**c**) independent experiments. Source data are provided as a Source Data file

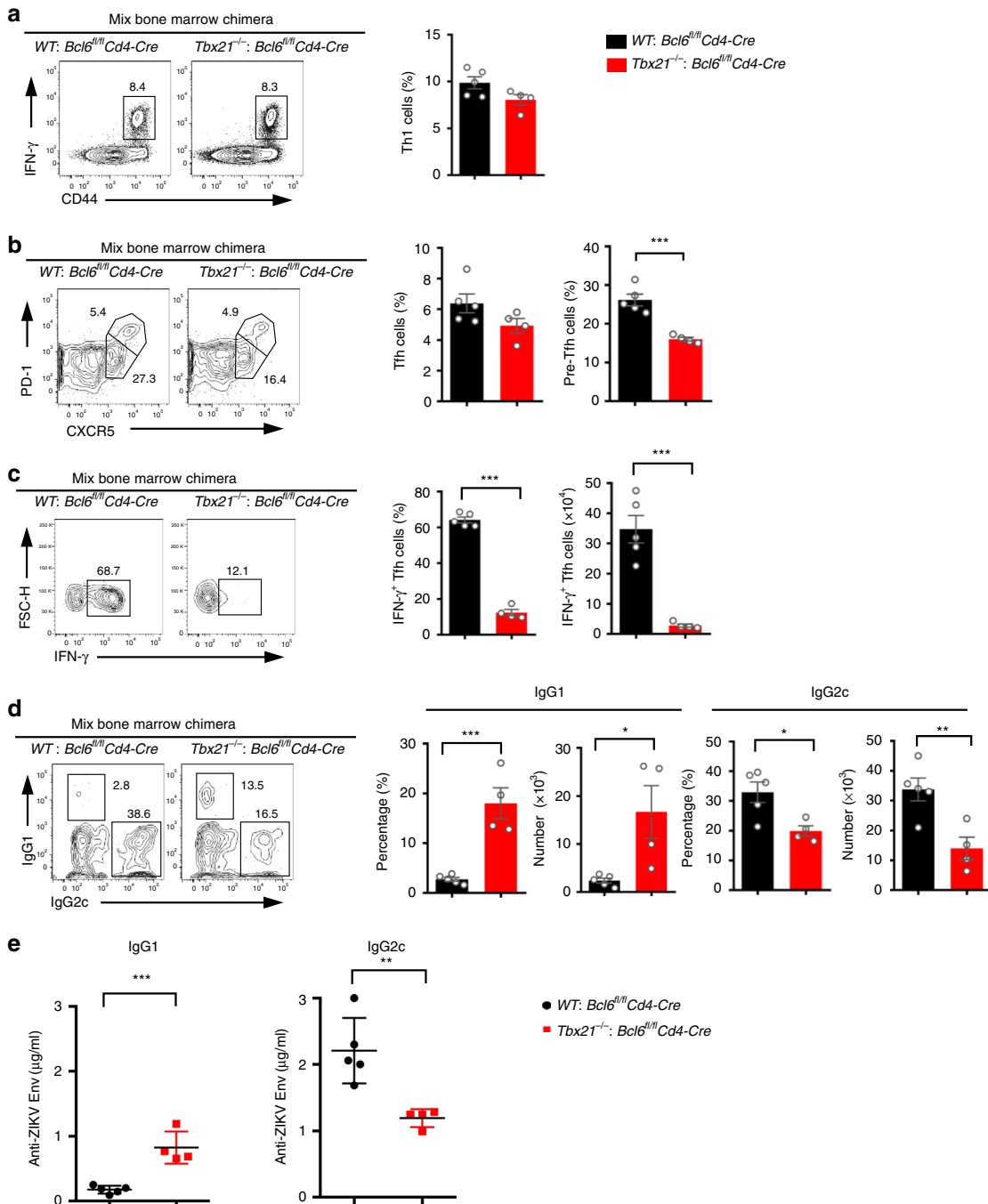

**Fig. 6** Th1-like Tfh cell is required for IgG2c antibody class switching. Mixed bone marrow chimeras (*WT: Bcl6^{fl/fl}Cd4*-Cre; and *Tbx21^{−/−}: Bcl6^{fl/fl}Cd4*-Cre) were administered with anti-IFNAR1 antibody one day prior to ZIKV infection. Splenocytes and sera were collected on 14 dpi for measurement of Tfh cell and antibody responses, respectively ($n = 5$ for *WT: Bcl6^{fl/fl}Cd4*-Cre group and $n = 4$ for *Tbx21^{−/−}: Bcl6^{fl/fl}Cd4*-Cre group). **a** Representative flow cytometry plots of IFN-$\gamma^+$ cells in CD4$^+$CXCR5$^-$ T cells were presented as Th1 cells (left panel); bar graphs summarized the percentages of Th1 cells in each group (right panel). **b** Representative flow cytometry plots of CXCR5$^{high}$PD-1$^{high}$ cells and CXCR5$^{medium}$PD-1$^{medium}$ in CD4$^+$CD44$^{high}$CD62L$^{low}$ T cells were presented as Tfh cells and pre-Tfh cells (left panel); bar graphs summarized the percentages of cells in each group (right panel). **c** Representative flow cytometry plots of IFN-$\gamma$ staining in CXCR5$^{high}$PD-1$^{high}$ Tfh cells (left panel); the percentages and numbers of IFN-$\gamma^+$ Tfh cells were summarized by bar graphs (right panel). **d** Representative flow cytometry plots of IgG1 and IgG2c intracellular staining in B220$^{low}$CD138$^+$IgD$^{low}$ cells (Left panel); bar graphs summarized the percentages and numbers of IgG1$^+$ and IgG2c$^+$ cells (right panel). **e** The concentrations of ZIKV envelope specific IgG1 and IgG2c antibodies in chimeras were measured by ELISA. The summary data were presented as mean ± SEM. Statistical differences were determined by Student's *t* test and *p* values were indicated by *$p < 0.05$, or **$p < 0.01$, or ***$p < 0.001$. Data (**a**–**e**) are pooled from two independent technical duplicates. Source data are provided as a Source Data file

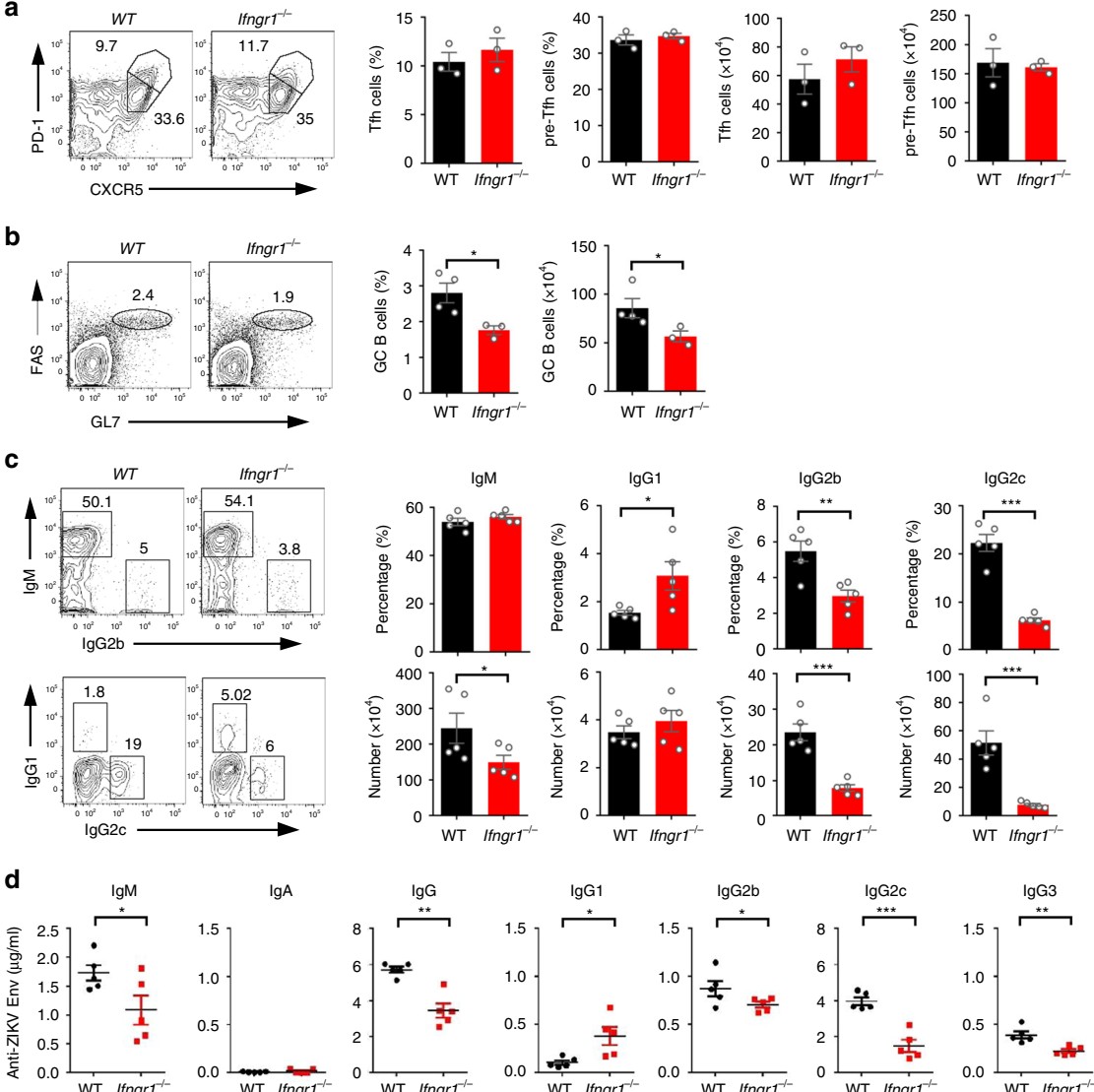

**Fig. 7** The IFN-γ pathway is required for IgG2c antibody class switching. **a–d** WT mice and *Ifngr1−/−* mice were infected by ZIKV with anti-IFNAR1 blocking antibody at one day prior to infection. Splenocytes were collected for measurement of Tfh and GC B cell responses on 7 dpi and antibody-producing B-cell responses on 14 dpi; sera were collected on 14 dpi for assessment of antibody responses (*n* = 3–5 for each group). **a** Representative flow cytometry plots of CXCR5highPD-1high cells and CXCR5mediumPD-1medium cells in CD4+CD44highCD62Llow cells were presented as Tfh cells and pre-Tfh cells (left panel); bar graphs summarized the percentages and numbers of cells in each group. **b** Representative flow cytometry plots of FAS+GL7+ cells in B220+ B cells were presented as GC B cells (left panel); bar graphs summarized the percentages and numbers of GC B cells in each group (right panel). **c** Representative flow cytometry plots of IgM, IgG1, IgG2b, and IgG2c staining of IgDlow B cells (left panel); bar graphs summarized the percentages and numbers of cells (right panel). **d** The concentrations of ZIKV envelope specific IgM, IgA, total IgG, IgG1, IgG2b, IgG2c, and IgG3 antibodies in sera were determined by ELISA. The summary data were presented as mean ± SEM. Statistical differences were determined by Student's *t* test and *p* values were indicated by *$p < 0.05$, or **$p < 0.01$, or ***$p < 0.001$. Data shown represent two (**a–d**) independent experiments. Source data are provided as a Source Data file

elicitation of ZIKV-specific neutralizing antibodies and long-term maintenance of antibody responses (Fig. 4). The differences between our data and Miyauchi's data[11] may be due to either the engagement of different antigen-presenting cells by inactivated influenza virus and viable ZIKV, or the divergent micro-milieu in which Th1, Tfh, and B cells interact. Further investigation into the regional immune responses should offer solution to this discrepancy.

Cytokines play vital roles in antibody class switching. IFN-γ is critical for the production of IgG2c in C57BL/6 mice (or IgG2a in BALB/c mice), whereas IL-4 contributes to the generation of IgG1 antibody. It has been reported that Tfh cells

constitute the main producer of IL-4 during antibody production stage[25], indicating Tfh cells are responsible for IgG1 class switching[21,26]. Likewise, IFN-γ-expressing cells, usually Th1 cells, are thought to be the mediator of IgG2c class switching. In vitro, co-culture Th1 cells with B cells can induce IgG2a production[20], suggesting Th1 cells have the potential to mediate IgG2c class switching. Indeed, a recent study showed that during vaccination with an inactivated influenza A virus, a sustainable IgG2c response could be generated in the absence of Tfh cells and GC B cells, but depend on the IFN-γ-expressing Th1 cells that are located at the T-B border[11]. In contrast, we found that IFN-γ-expressing Th1-like Tfh cells are

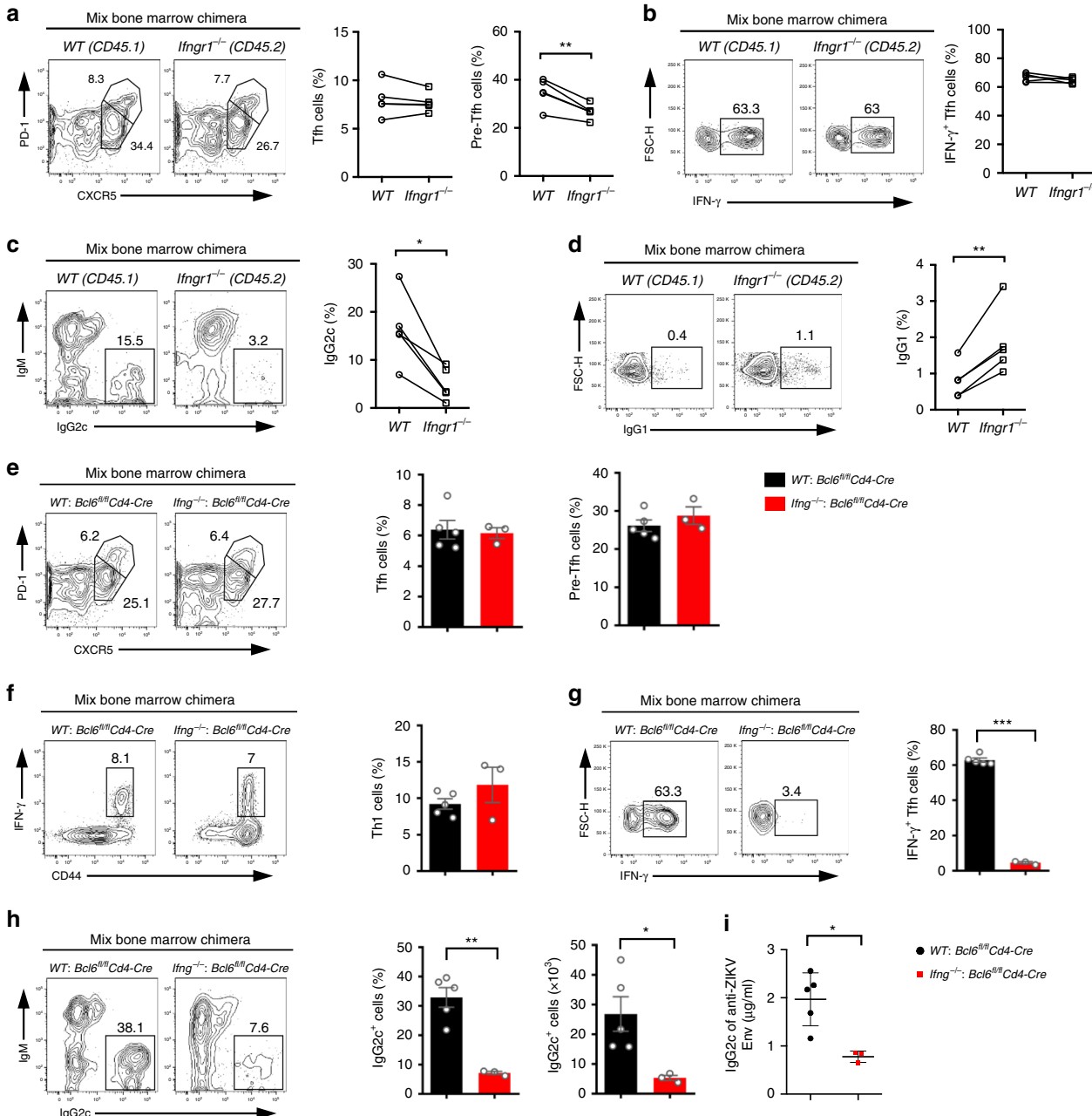

**Fig. 8** IFN-γ produced by Th1-like Tfh cells is required in a B cell-intrinsic manner for IgG2c antibody class switching. **a–d** CD45.1+ WT: Ifngr1−/− mixed bone marrow chimeras were administered with anti-IFNAR1 antibody 1 day prior to ZIKV infection. Splenocytes were collected on 14 dpi for measurement of Tfh cell and antibody responses (n = 5). **a** Representative flow cytometry plots of CXCR5highPD-1high cells and CXCR5mediumPD-1medium cells were presented as Tfh cells and pre-Tfh cells (Left panel); summary of the percentages of these cells (right panel). **b** Representative flow cytometry data of IFN-γ staining in splenic WT (CD45.1+) Tfh cells and Ifngr1−/− (CD45.2+) Tfh cells at day 14 after ZIKV infection (left panel); summary of the percentages of WT or Ifngr1−/− IFN-γ+ Tfh cells (Right panel). **c, d** Representative flow cytometry data of IgG2c (**c**) and IgG1 (**d**) intracellular staining in B220lowCD138+IgDlow cells (left panel); summary of the percentages of these cells (Right panel). **e–i** WT: Bcl6fl/flCd4-Cre chimeras, and Ifng−/−: Bcl6fl/flCd4-Cre chimeras were administered with anti-IFNAR1 antibody one day prior to ZIKV infection (n = 5 for WT: Bcl6fl/flCd4-Cre group and n = 3 for Ifng−/−: Bcl6fl/flCd4-Cre group). **e** Representative flow cytometry plots of CXCR5highPD-1high cells and CXCR5mediumPD-1medium cells were presented as Tfh cells and pre-Tfh cells (left panel); bar graphs summarized the percentages of these cells (right panel). **f** Representative flow cytometry plots of IFN-γ+ cells in CD4+CXCR5− T cells were presented as Th1 cells (left panel); bar graphs summarized the percentages of Th1 cells (right panel). **g** Representative flow cytometry plots of IFN-γ staining in CXCR5highPD-1high Tfh cells (left panel); the percentages of IFN-γ+ Tfh cells were summarized by bar graphs (right panel). **h** Representative flow cytometry data of IgG2c intracellular staining in B220lowCD138+IgDlow splenocytes (left panel); bar graphs summarized the percentages and numbers of IgG2c+ cells (right panel). **i** The concentrations of ZIKV envelope specific IgG2c antibodies in chimeras were measured by ELISA. The summary data were presented as mean ± SEM. Statistical differences were determined by Student's t test and p values were indicated by *p < 0.05, or **p < 0.01, or ***p < 0.001. Data (**a–i**) are pooled from two independent technical duplicates. Source data are provided as a Source Data file

the main contributor to antigen specific IgG2c class switching in ZIKV infection. The production of antigen specific IgG2c antibody was dramatically reduced in the absence of Th1-like Tfh cells in ZIKV infected mice that had been genetically knocked-out of transcription factor Bcl6 or T-bet. However, we cannot distinguish whether Th1-like Tfh cells or Th1-like pre-Tfh cells mediated the IgG2c antibody class switching in our current experimental system, because both of them disappeared in the absence of Bcl6 and T-bet. Previous study showed that IFN-γ-expressing helper T cells, conjugating B cells residing in the T-B border, can lead to IgG2c class switching[21]. Therefore, it is likely that Th1-like pre-Tfh cells mediate the IgG2c antibody class switching in the T-B border, whereas Th1-like Tfh cells play an important role in the expansion of IgG2c-producing cells.

In our study, ZIKV acts as a natural live attenuated vaccine for normal immunocompetent mice due to its restricted replication. Immunization with this form of attenuated vaccine induced very weak Tfh cell and antibody responses (Fig. 1; Fig. 2a and Supplementary Fig. 3). Interestingly, immune-modulated ZIKV vaccination (by pre-treatment with antibody to IFNAR1) greatly enhances the induction of protective neutralizing antibodies. Such an effect may be explained by the fact that immuno-modulation transiently allows ZIKV replication and results in more antigens being produced and consequently presented by antigen-presenting cells to facilitate Tfh cell development and antibody induction. This is accompanied by simultaneous induction of more pro-inflammatory factors that boost the overall immune responses. A similar scenario may occur in many attenuated or inactivated human vaccines that do not have sufficient immunogenicity; for such vaccines, immuno-modulation strategy could potentially augment their immunogenicity, and thus our findings may have important implications for improving vaccine potency.

In summary, our study has revealed an immuno-modulation strategy by which the elicitation of protective antibody responses could be greatly enhanced after ZIKV infection in mice. We also defined the characteristics and functions of a new subset of Tfh cells, Th1-like Tfh cells, in this model. Our study may provide a useful method for manipulating humoral responses in vaccine development.

## Methods

**Ethics statement**. All mice were bred and maintained in specific pathogen-free barrier facilities and used at 6–10 weeks of age. All mouse experiments were performed strictly according to rules of care and use of laboratory animals established by Institutional Animal Care and Use Committee of the Institut Pasteur of Shanghai, Chinese Academy of Sciences (Approval number: A2018027). For virus infection experiments, all procedures were completed under BSL-2 and ABSL-2 laboratory conditions as approved by the Biosafety Committee of Institut Pasteur of Shanghai.

**Cells and virus**. C6/36 mosquito cells (ATCC; Cat. No. CRL-1660) and Vero cells (ATCC; Cat. No. CRL-1586) were cultured in MEM (Gibco; Cat. No. 11095080) and DMEM (Gibco; Cat. No. 11965092) with 10% FBS (Gibco; Cat. No. 10099141). Zika virus (ZIKV), SZ-WIV01 strain, was kindly provided by Wuhan Institute of Virology, Chinese Academy of Sciences and propagated in C6/36 cells. Amplified ZIKV was aliquoted into 2 ml tubes and titrated on Vero cell monolayer by 10-fold serial dilutions, and recorded as plaque-forming units (PFUs). All virus stock was kept at −80 °C until use.

**Plaque assay and PRNT**. Vero cells were plated at $1 \times 10^5$ cells/well in a 24-well tissue culture plate and incubated overnight at 37 °C, 5% CO₂. At day 2, cells were washed with warm DMEM before virus infection. Virus samples were serially diluted in DMEM supplemented with 1% penicillin/streptomycin (P/S) and then incubated with Vero cells for 2 h for virus adsorption. The viral inoculum was removed and DMEM supplemented with 1.5% fetal bovine serum (FBS), 1% P/S, 1% carboxymethylcellulose (CMC) was added. Vero cells were incubated with the CMC overlay for 4 days, and then the overlay was removed. Cells were fixed with 4% paraformaldehyde (PFA) for 4 h, then stained with 2.5% Crystal Violet solution

for plaque counting. For plaque reduction neutralization test (PRNT), Zika virus (50–100 PFU/well) was mixed with serial diluted inactivated serum at a volume ratio of 1:1. Mixed Virus-Ab complex was incubated at 37 °C, 5% CO₂ for 1 h, followed by standard plaque assay procedures.

**Mice**. C57BL/6 and BALB/C mice were purchased from Beijing Vital River Laboratory Animal Technology (licensed from Charles River). $Bcl6^{fl/fl}$, $Tbx21^{-/-}$, and $Cd4$-Cre mice were purchased from Jackson Laboratories. $Bcl6^{fl/fl}$ mice were bred with $Cd4$-Cre mice to generate $Bcl6^{fl/fl}Cd4$-Cre mice. IFN-γ-YFP reporter mice were provided by Dr. Z.N. Yin (Jinan University, Guangzhou, China). CXCR5-GFP reporter mice, $Tbx21^{fl/fl}Cd4$-Cre mice, and $Ifng^{-/-}$ mice were provided by Dr. L.L. Ye (Third Military Medical University, Chongqing, China). $Ifngr1^{-/-}$ mice were provided by Dr. X.Z. Liang (Institut Pasteur of Shanghai, Shanghai, China).

**Construction of mixed bone marrow chimeras**. Bone marrow cells from either $Cd45.2$ WT and $Bcl6^{fl/fl}Cd4$-Cre, or $Tbx21^{-/-}$ and $Bcl6^{fl/fl}Cd4$-Cre, or $Ifng^{-/-}$ and $Bcl6^{fl/fl}Cd4$-Cre mice were mixed at a ratio of 1:1, the mixed bone marrow were then transferred to the irradiated $Tcrb^{-/-}$ recipient mice to create three different types of chimeric mice. Bone marrow cells from $Cd45.1$ WT and $Ifngr1^{-/-}$ were mixed at a ratio of 1:1, then transferred to the irradiated $Rag1^{-/-}$ recipient mice to create the fourth type of chimera. After 6 weeks' reconstitution, chimeras were used to evaluate the immune response to ZIKV infection.

**Mouse infection, immunization, and sample collection**. For infection experiments, mice were injected intraperitoneally (I.P.) with 1 mg interferon α/β receptor (IFNAR1) blocking monoclonal antibody MAb-5A3 (BioXcell) or control antibody at 1 day prior to infection (day 0). On day 1, mice were infected with $10^5$–$10^6$ PFUs of ZIKV by subcutaneous (S.C) route. Lymphocytic choriomeningitis virus (LCMV) (Armstrong strain) was used to infect mice at $2 \times 10^5$ PFUs through I.P. route. For protein immunization experiment, recombinant ZIKV-E80 (Ectodomain of the envelope protein of ZIKV) or NP-OVA emulsified in Alum adjuvant (Alhydorgel) were administered via S.C route to mice with or without MAb-5A3 antibody pre-treatment. Mouse serum was collected and inactivated at 56 °C, 30 min, then stored at −80 °C until use. After experiments, mice were euthanized at different time points, splenocytes and draining lymph nodes were collected and used for flow cytometry analysis.

**Flow cytometry**. The antibodies used for flow cytometry were listed in Supplementary Table 1. As preciously described[27], all the surface staining was done on ice for 0.5 h except for CXCR5 and IgG2b staining, for which anti-CXCR5-Biotin or anti-IgG2b-Biotin was stained for 1 h, followed by staining with either PE or Brilliant Violet 421-labeled streptavidin on ice for 0.5 h. For intracellular staining of T-bet, Bcl6, and Eomes, cells were fixed with 3.7% formaldehyde for 15 min at room temperature after staining of surface markers and permeabilized with 0.2% Triton X-100 in PBS for 20 min at room temperature, and then stained for 1 h at room temperature. After staining, cells were washed three times with 0.01% Triton X-100 in PBS. For intracellular staining of cytokines, cells were stimulated with 10 ng/mL phorbol 12-myristate 13-acetate (PMA) and 1 μg/mL ionomycin for 4 h, and 2 μg/mL Brefeldin A was added for the last 2 h of stimulation. Cells were fixed with 3.7% formaldehyde for 15 min at room temperature after staining of surface markers and permeabilized with 0.2% saponin in PBS on ice for 20 min before intracellular staining. For IL-21 staining, IL-21R-Fc chimera protein was added for 1 h at room temperature, followed by staining with AF647-labeled anti-Human IgG on ice for 0.5 h. After staining, cells were washed twice with 0.2% saponin in PBS to reduce the background.

**RNA-seq analysis**. After ZIKV infection, CD4⁺ T cells in spleens and draining lymph nodes in IFN-γ-YFP reporter mice were enriched with negative selection by using biotin-labeled CD8a, TER-119, and B220 antibody (Supplementary Table 1) and streptavidin magnetic beads (Cat. No., SVM-10-10, Spherotech) before surface markers staining. CD4⁺CD44^high CD62L^low cells were sorted by FACS Aria II cell sorter (BD Bioscience) into three populations (CXCR5⁻CXCR3⁺YFP⁺, CXCR5^high PD-1^high CXCR3⁺YFP⁺ and CXCR5^high PD-1^high CXCR3⁻YFP⁻ cells), and then resuspended in Trizol for RNA extraction. Two replicates of each cell subset were obtained. RNA quality was assessed and sequenced using illumina Hiseq X Ten platform in Novogene. Raw reads were aligned to the mm10 mouse genome using HISAT2 version 2.0.5 and counts of transcripts were calculated using htseq-count. Differential analysis between cell types (Th1-like Tfh, Tfh, and Th1 cells) was done by a count-based method limma, which was implemented in R, and mean-variance modeling at the observational level (voom) was used for normalization[28,29]. Paired sample analysis was employed to compensate for batch effect. Significantly expressed genes were first screened by BH-adjusted p value 0.05[30], and further filtered with 1.5-fold change and threefold change for Th1-like Tfh cells vs. Tfh cells and Th1 cells vs. Tfh cells, respectively. For heatmap, log2 scale of the FPKM values (fragments per kilobase of exon per million fragments mapped reads) of the significant genes (including up- and downregulated genes resulted from comparisons between Th1-like Tfh cells and Tfh cells, Th1-like Tfh cells and

Th1 cells, as well as Tfh cells and Th1 cells) were used as input for heatmap generation. Biclustering (both genes and samples were clustered) was done for a global view of all the significantly changed genes across all cell types through R package heatmap3. Enrichment analysis was performed using GSEA software v3.0 (Broad Institute) mode. To be consistent with the method for identifying significant genes, we used the $t$-statistic output from limma as a metrics for ranking. A set of 1000 gene permutations was set as default, and gene sets were obtained through collecting pathways from KEGG and Biocarta, as well as biological processes from GO. A gene set with a $q$ value of false discover rate (FDR) <0.05 was considered as significantly enriched. The input data of cpm in scatter plot was also normalized.

**Immunoglobulin isotyping**. The assay was performed according to protocol provided by the Immunoglobulin Isotyping Kit (Southern Biotech). Briefly, purified ZIKV envelope protein or capture antibody dissolved in PBS was coated onto a 96-well flat-bottom plate (Costar) overnight. The liquid content on plates were decanted, and then the plates were blocked with PBS containing 3% bovine serum albumin (BSA) and 0.05% Tween 20 (PBST) for 2 h at RT. An 1:100 diluted serum solutions, or different class and isotype of murine Ig standard diluted in 1% BSA PBST, were then added in duplicated wells and incubated for 1 h at RT. PBST with 1% BSA and 1:500 diluted horseradish peroxidase (HRP)-conjugated anti-mouse different class and isotype Ig antibodies were added, and then incubated for 1 h at RT. Color development was performed using 3,3′, 5,5-tetramethylbenzidine (TMB) One Solution (Life Technology), with 1 mM HCl ending the reaction, followed by colorimetric analysis at 450 nm in a 96-well plate reader (Thermo Fisher Scientific).

**In vivo protection experiment**. For each experimental group, 200 μl of pooled sera were adoptively transferred to immunocompromised AG6 mice (Type I and Type II interferon receptor double KO) by I.P route at day 0. At day 1, mice were infected with $10^3$ PFU Zika virus by S.C route. Body weight and survival rate were monitored from day 1 to 15.

**Statistical analysis**. $PRNT_{50}$ titers, IgG isotype titers, T cell responses, and germinal center (GC) B responses were compared for statistical differences by one-tailed Student's $t$ test. $PRNT_{50}$ values were obtained through PROBIT regression analysis (SPSS Statistics). The survival curves were analyzed by log-rank test, and the body weight was analyzed by two-way ANOVA (Turkey correction). Statistical significance levels were reported as the following: $*p < 0.05$; $**p < 0.01$; $***p < 0.001$ or less.

**Reporting summary**. Further information on research design is available in the Nature Research Reporting Summary linked to this article.

## Data availability
The authors declare that the data supporting the findings of this study are available within the article and its supplementary information files, or are available upon request to the corresponding authors. The RNA-seq data that support the findings of this study have been deposited in Gene Expression Omibus with the primary accession code GSE121250. Source data are provided as a source data file.

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

## Acknowledgements
We thank Dr. H. Tang for kindly providing NKG2D antibody, and Dr. X.Z. Liang for kindly providing $Ifngr1^{-/-}$ mice. This work was supported by the following grants: Strategic Priority Research Program of the Chinese Academy of Sciences (XDB29040301, X.J.; XDB29030103, H.W.), National Key R&D Program of China (2016YFA0502202, H.W., L.Y.; 2016YFC1201000, X.J), Ministry of Science and Technology of China (2016YFE0133500), European Union Horison 2020 Research (X.J., J.Z.) and Innovation Programme under ZIKAlliance Grant Agreement 734548 (X.J., H.W., and J.Z.), the National Natural Science Foundation of China (31570886, H.W.), and the National Sciences and Technology Major Projects for 'Major New Drugs Innovation and Development' (2018ZX09711003-001-004, Y.X.).

## Author contributions

X.J. and H.W. conceived and supervised the research; H.L., J.T., Z.L., L.Y., X.J. and H.W. contributed to the design of the project and discussions; H.L., J.T. and Z.L. performed the experiments; Y.H. helped with the mixed bone marrow chimeras experiments; Y.L. and P.H. did the bioinformatics analysis; Y.X., L.Y., Z.Y. and J.Z. assisted with mouse models; H.L., J.T., Z.L., X.J. and H.W. wrote the paper.

## Additional information

**Competing interests:** The authors declare no competing interests.

