## [Peer Review File · Nature Communications]

Reviewers' Comments:

Reviewer #1:

Remarks to the Author:

This study described "Th1-like" Tfh cells in the context of Zika virus infection. The study is well done on technical side, using various genetically modified models to demonstrate that IFN γ -produced cells constitute a large portion of Tfh cells, that generation of these cells depend on Bcl6 and T-bet, and the IFN γ produced is important for IgG2 class switching. However, on conceptual side, there is little new. It has been already known that Tfh cells produce IFN γ , which requires Tbet, and generation of Tfh cells depends on Bcl6, although the portion of IFN γ -Tfh cells might be higher in frequency in ZIKV infection compared with LCMV infection.

The authors may have different views of Tfh differentiation process. Generally PD1-CXCR5+ cells are committed Tfh cells, not pre-Tfh cells, PD1+CXCR5+ cells are GC Tfh cells, and the interconversion between these two subsets can be quite dynamic. The data actually showed that both subsets can produce IFN γ .

The authors focused on a specific role of IgG2c class-switch. Based on the data using CD4Cre+Tbet fl/fl and Ifngr1 KO mice, both IgG2b and 2c were affected. The germline deletion of Tbet had a more specific effect on IgG2c. These need to be reconciled.

Reviewer #2:

Remarks to the Author:

The authors report that Th1-like Tfh cells, which co-express transcription factors T-bet and Bcl6 and produce IFN γ , are required to induce protective IgG2 antibody responses against Zika virus (ZIKV) infection. Overall, this manuscript reports an important study of the T cell requirements for inducing protective antibody responses against a clinically relevant virus. However, the study could be strengthened by the use of cleaner models to pinpoint the role of Th1-like Tfh cells and where IFN γ signaling is required. Also, this work represents only a minor advance from previously reported work on the role of Tbet in Tfh function (ref 15) and it is not clear how distinct these "Th1-like Tfh" cells are from "ex-Tfh1" cells previously reported (Ref 14).

They perform their experiments in immunocompetent mice, which are normally resistant to ZIKV infection due to type I IFN responses, and therefore rely on a model of anti-IFNAR1 blockade prior to ZIKV infection. The authors nicely argue that anti-IFNAR1 is not responsible for the Th1-like Tfh cells they observe in ZIKV infection by performing parallel experiments with LCMV infection with and without anti-IFNAR1 pre-treatment. They show that LCMV infection, which can occur in mice with intact type I IFN responses, induces similar degrees of total Tfh cells and IFN γ + Tfh cells regardless of anti-IFNAR1 treatment.

The experiments depleting Th1-like Tfh cells using total or T cell-specific Tbx21 deficiency are the main weakness of this paper. While the authors intend to use these mice to specifically deplete Th1-like Tfh cells, they find that Tbx21^{-/-} mice have decreased numbers of total pre-Tfh cells (Fig. 5a) and Tbx21f/fCD4-Cre mice have decreased numbers of both pre-Tfh cells and Tfh cells (Fig. S7a). Therefore, it is unclear whether the decreased IgG2c antibody titers are due to specific loss of Th1-like Tfh cells or just overall reduction of Tfh cells. The decrease in pre-Tfh and Tfh cell counts is potentially not an issue if the loss of Th1-like Tfh cells is solely responsible for the decreased counts – the authors could see if non-Th1-like Tfh cell counts remain the same in Tbx21-deficient mice. However, if both Th1-like and non-Th1-like Tfh cell counts are reduced, the authors would need to use a cleaner

experimental model (e.g. Bcl6f/fCD4-Cre:Tbx21-/- mixed bone marrow chimera mice). The authors could also demonstrate that IFN γ is the operative signal from Th1-like Tfh cells that induces IgG2 switching with the use of Bcl6f/fCD4-Cre:Ifng-/- mixed bone marrow chimera mice.

Additionally, Figure 6 argues that IFN γ signaling is required for class switching to IgG2, but does not pinpoint which cell type requires such signaling due to the use of total knockout Ifngr1-/- mice. The authors do demonstrate unimpaired pre-Tfh and Tfh cell induction in Ifngr1-/- mice, but do not demonstrate their cytokine profile – does Ifngr1 deficiency affect the ability of Tfh cells to produce IFN γ ? Or is Ifngr1 deficiency in B cells solely responsible for decreased IgG2 production? The authors could add clarity to this part of the manuscript by limiting Ifngr1 deficiency to specific cell types.

Minor comments

1. The manuscript contains a number of typos throughout.
2. The initial characterization of Tfh cell response and cytokine profile is performed in BALB/c mice (Fig. 1-2). However, later studies are done in Bcl6f/fCD4-Cre mice on C57BL/6 background. Do the authors have parallel characterizations in both strains? Obviously these two strains have very different T cell differentiation proclivity.
3. Statistics are missing in most of the subfigures within Fig. 1.
4. In Figure 2b, please indicate on which day Tfh cells were assessed for IFN γ production.
5. Line 213: Gata3 is not a conventional Tfh-associated gene.

Reviewer #3:

Remarks to the Author:

This manuscript by Liang and colleagues report a role for Th1-like Tfh in shaping the humoral response to Zika virus (ZIKV) infection in a mouse model. The authors used BALB/c mice with IFNAR inhibition to enable productive ZIKV infection as evidenced by detectable viremia to identify how this subset of T cells is important in inducing robust IgG response, especially in the generation of IgG2 antibodies. The authors suggest that their findings provide insights into the targeting of Th1-like Tfh cells for manipulation to induce robust humoral response to vaccination.

The findings are indeed interesting and the authors appear to have done extensive amount of work to pinpoint an important role for this subset of Tfh cells. I appreciated the attention to detail in ruling out a possible effect of IFNAR blocking antibodies in shaping the humoral response. Likewise, the use of relevant knockout mouse models supported the line of thinking clearly. I have a few comments for the authors to consider.

1. The authors hint at an underlying factor that could account for the difference in humoral response to ZIKV infection in IFNAR blocked vs control BALB/c mice. Viremic infection in the former likely have resulted in more antigens being expressed compared to the latter. Greater antigen load could hence have resulted in more antigen presentation by Tfh cells or activation of these cells in the lymph nodes, or both. Alternatively, the immune response could have been dependent on replicating ZIKV, where the pro-inflammatory factors expressed in infected cells could have stimulated antigen presenting cells and/or T cells in ways that cannot be reproduced even if an equivalent level of inactivated antigens were inoculated in these animals. The inclusion of an inactivated ZIKV control in the experiments could have clarified these questions. It would also inform on whether, besides the degree of cellular immunity, qualitatively different humoral response could be elicited by replicating compared to non-replicating vaccines due to the differences in Th1-like Tfh stimulation. This possibility could have important implications on how the pipeline of candidate Zika vaccines should be prioritised for clinical development. Perhaps the authors would consider adding a discussion on this issue?

2. Line 441 indicated that 500-1000 pfu/ml virus was incubated with an equivalent volume of serially diluted serum before inoculation onto a monolayer of Vero cells in a 24-well plate. Unless the volume was very small, which could introduce inaccuracies, the number of pfu appears to be very large for a 24-well plate assay. Is this description correct? Please clarify.

3. Figure 1a suggests that there is some background ZIKV neutralization activity in the uninfected animal controls (a 50% neutralisation titer of nearly 2 logs at 14 days). This is rather unusual. Can the authors explain what is going on?

4. Line 155. Figure 2b should be labelled here as Figure 3b.

5. There are scattered typographical/grammatical errors in the manuscript.

Point-by-point response to reviewers' comments

Editorial comments:

In this case reviewers raised significant concerns regarding some aspects of the models you present and suggest (1) further in vivo work to further enhance your manuscript, to fully and more appropriately understand the role of Th1 like Tfh cells in this context, which we agree would be beneficial in this case. Reviewers additionally raised concerns regarding (2) the molecular mechanisms at play and we agree further elucidation in this context would be necessary here. That is not to say we find any point raised by reviewers as less important and strongly advise you to address each point of concern raised by each reviewer as fully as possible by the most appropriate means.

Response:

We interpret the above as that you agree with the reviewers on two of their major concerns and you suggest that we address these issues by (1) performing more *in vivo* work, and (2) elucidating relevant molecule mechanisms.

To address these important issues, we took the sensible suggestion from reviewer #2 of doing mixed bone marrow chimera experiments. We performed new experiments by giving *Tcrb*^{-/-} mice two different mixture of cells: an equal number of *WT: Bcl6^{fl/fl}Cd4-Cre*, and *Tbx21^{-/-}: Bcl6^{fl/fl}Cd4-Cre*, to construct mixed bone marrow chimeras. As expected, there are specific deficiency in Th1-like Tfh cells, but still capable of generating conventional Tfh cells, with about 40% reduction of pre-Tfh cells in these mixed bone marrow chimera (**Reb. Figure 1a, 1b**), which is consistent with that observed in *Tbx21^{-/-}* mice (Fig. 5a). And IgG2c production is impaired with IgG1 production increased (**Reb. Figure 1c, 1d**). These new data support the conclusion that Th1-like Tfh (and -pre-Tfh), but not conventional Tfh cells, are responsible for IgG2c class switching in ZIKV infection. Reb. Fig.1a has been included in the revised paper as supplementary Fig.7c; Reb. Fig.1b, 1c, 1d have been included as revised Fig.5e, 5f, 5g and supplementary Fig. 7d.

Reb. Figure 1. Mixed bone marrow chimera experiments demonstrate that Th1-like Tfh cells are responsible for IgG2c class switching. Mixed bone marrow chimeras (*WT: Bcl6^{fl/fl}Cd4-Cre*; and *Tbx21^{-/-}: Bcl6^{fl/fl}Cd4-Cre*) were administered with anti-IFNAR1 antibody one day prior to ZIKV infection. Splenocytes and sera were collected at 14 dpi for measurement of Tfh cell and antibody responses, respectively (n=5 for *WT: Bcl6^{fl/fl}Cd4-Cre*, and n=4 for *Tbx21^{-/-}: Bcl6^{fl/fl}Cd4-Cre*). **(a)** Representative flow cytometry plots of CXCR5^{high}PD-1^{high} Tfh cells and CXCR5^{medium}PD-1^{medium} pre-Tfh cells (left panel); the percentages were

summarized in bar graphs (right panel). **(b)** Representative flow cytometry plots of IFN- γ staining in CXCR5^{high}PD-1^{high} Tfh cells (left panel); the percentages and numbers of IFN- γ ⁺ Tfh cells were summarized in bar graphs (right panel). **(c)** Representative flow cytometry plots of IgG1 and IgG2c intracellular staining in B220^{low}CD138⁺IgD^{low} cells (left panel); bar graphs summarized the numbers of IgG1⁺ and IgG2c⁺ cells (right panel). **(d)** The concentrations of ZIKV envelope specific IgG1 and IgG2c antibodies in chimera were measured by ELISA. The summary data were presented as mean \pm SEM. Statistical differences were determined by Student's t test and p values were indicated by * (p<0.05), or ** (p<0.01), or *** (p<0.001).

To study molecular mechanisms in greater depth, we again performed new experiments by giving *Rag1*^{-/-} mice an equal number of cells from *CD45.1: Ifngr1*^{-/-} to make a new type of mixed bone marrow chimera. These chimeric mice allowed dissection of whether IFN- γ signaling is required and by what cell types. In these chimeras, IFN- γ receptor-deficient Tfh cells developed normally, with slight reduction of pre-Tfh (**Reb. Figure 2a**), and these cells could produce similar levels of IFN- γ , compared to CD45.1⁺ wild type Tfh cells in the same chimeras (**Reb. Figure 2b**), consistent with that in *Ifngr1*^{-/-} mice (Supplementary Fig. 8a). during ZIKV infection. However, in the chimera, IFN- γ receptor-deficient B cells produced much reduced level of IgG2c antibodies than CD45.1⁺ wild type B cells (**Reb. Figure 2c**), suggesting that IFN- γ signaling is required for B cell to mediate antibody class switching. Reb. Figure 2a has been included as revised supplementary Figure 8c; Reb. Figure 2b, 2c have been included as revised Figure 6e, 6f.

Reb. Figure 2. IFN- γ signaling is required for B cell to mediate antibody class switching. *CD45.1*⁺ WT: *Ifngr1*^{-/-} mixed bone marrow chimeras were administered with anti-IFNAR1 antibody one day prior to ZIKV infection. **(a)** Representative flow cytometry plots of the reconstitution of CD45.2⁺ (*Ifngr1*^{-/-}) and CD45.2⁻ (*Cd45.1*) in CD4⁺ cells; Representative flow cytometry plots of Tfh cells and pre-Tfh cells of CD4⁺ CD44^{high} CD62L^{low} cells (Left panel); bar graphs summarized the percentages (Right panel). **(b)** Representative flow cytometry plots of IFN- γ production in Tfh cells from CD45.2⁺ (*Ifngr1*^{-/-}) and CD45.2⁻ (*Cd45.1* WT) mice (Left panel); bar graphs summarized the percentages (Right panel). **(c)** Representative flow cytometry plots of IgG2c production in B220^{low} CD138⁺ IgD⁻ cells from CD45.2⁺ (*Ifngr1*^{-/-}) and CD45.2⁻ (*Cd45.1* WT) (Left panel); bar graphs summarized the percentages (Right panel). The summary data were presented as mean \pm SEM. Statistical differences were determined by Student's t test and p values were indicated by * (p<0.05).

In addition to Th1-like Tfh cells, many other cell types such as NK, Th1 and CD8⁺ T cells can produce IFN- γ . To address the question of whether IFN- γ produced from Th1-like Tfh cells induces IgG2 switching, we created an *Ifng*^{-/-}: *Bcl6*^{fl/fl} *Cd4-Cre* mixed bone marrow chimera, in which Tfh cells all come from *Ifng*^{-/-} CD4⁺ T cells. As expected, these mixed bone marrow chimera could generate Tfh and pre-Tfh cells normally (**Reb. Figure 3a**), but could not generate IFN- γ ⁺ Th1-like Tfh cells, distinctly different from the WT: *Bcl6*^{fl/fl} *Cd4-Cre* mixed bone marrow chimera (**Reb. Figure 3b**). With the loss of IFN- γ ⁺ Th1-like Tfh cells, *Ifng*^{-/-}: *Bcl6*^{fl/fl} *Cd4-Cre* mixed bone marrow had fewer IgG2c producing cells and lower level of ZIKV envelope specific IgG2c in sera after

ZIKV infection (**Reb. Figure 3c, 3d**), suggesting IFN- γ from Th1-like Tfh cells is responsible for IgG2 switching. Reb. Figure 3a, 3b have been included in the revised paper as supplementary Figure 8d, 8e; Reb. Figure 3c, 3d have been included as revised Figure 6g, 6h.

Reb. Figure 3. IFN- γ from Th1-like Tfh cells is responsible for IgG2 switching. Mixed bone marrow chimeras (*WT: Bcl6^{fl/fl}Cd4-Cre*; and *Ifng^{-/-}: Bcl6^{fl/fl}Cd4-Cre*) were administered with anti-IFNAR1 antibody one day prior to ZIKV infection. Splenocytes and sera were collected at 14 dpi for measurement of Tfh cell and antibody responses, respectively (n=5 for *WT: Bcl6^{fl/fl}Cd4-Cre*, and n=3 for *Ifng^{-/-}: Bcl6^{fl/fl}Cd4-Cre*). **(a)** Representative flow cytometry plots of CXCR5^{high}PD-1^{high} Tfh cells and CXCR5^{medium}PD-1^{medium} pre-Tfh cells (left panel); the percentages were summarized in bar graphs (right panel). **(b)** Representative flow cytometry plots of IFN- γ staining in CXCR5^{high}PD-1^{high} Tfh cells (left panel); the percentages of IFN- γ ⁺ Tfh cells were summarized in bar graphs (right panel). **(c)** Representative flow cytometry plots of IgG2c intracellular staining in B220^{low}CD138⁺IgD^{low} cells (left panel); bar graphs summarized the numbers of IgG2c⁺ cells (right panel). **(d)** The concentrations of ZIKV envelope specific IgG2c antibody in chimeras were measured by ELISA. The summary data were presented as mean \pm SEM. Statistical differences were determined by Student's t test and p values were indicated by * (p<0.05), or ** (p<0.01), or *** (p<0.001).

Comments from Reviewer #1

(1) This study described “Th1-like” Tfh cells in the context of Zika virus infection. The study is well done on technical side, using various genetically modified models to demonstrate that IFN γ -produced cells constitute a large portion of Tfh cells, that generation of these cells depend on Bcl6 and T-bet, and the IFN γ produced is important for IgG2 class switching. However, on conceptual side, there is little new. It has been already known that Tfh cells produce IFN γ , which requires Tbet, and generation of Tfh cells depends on Bcl6, although the portion of IFN γ -Tfh cells might be higher in frequency in ZIKV infection compared with LCMV infection.

Response:

We thank the Reviewer #1 for acknowledging the technical strength of our paper, but would also like to highlight the conceptual advance our paper has been, albeit the degree of advancement may

be judged subjectively. The novelty of our results may be highlighted from the following three aspects.

It is true that Tfh cells produce IFN- γ , for which T-bet is required, and it is known that the generation of Tfh cells depends on Bcl6. However, neither of which has been established in a Zika virus infection model, not to mention in immunocompetent mice that are hard to infect by Zika virus. Moreover, most other studies of Tfh have not included parallel investigation of pre-Tfh and GC B responses. Together, the three population of cells give a more comprehensive overview of the complex interactions among different cell subsets.

Secondly, some previous papers have indicated the existence of IFN- γ -producing Th1-like Tfh cells, however, quantitatively, such a large expansion of Th1-like Tfh cells (about 70% of the total Tfh cells) has never been observed before. It would not be unreasonable to emphasize that quantity is often as important as quality in a biological process. The specificity of such a phenomenon, i.e., a greater expansion of Th1-like Tfh cells in ZIKV infection than that in LCMV (supplementary Fig.4b), Influenza and Dengue infections (unpublished data) also added another level of novelty.

Functionally, we showed that Th1-like Tfh cells are responsible for IgG2c class switching, thus providing a convincing piece of evidence in support of the idea that IFN- γ being critically important for class-switch recombination [1].

Based on the above, we think this study may have some degree of novelty.

Reference cited:

1. Higgins BW, McHeyzer-Williams LJ, McHeyzer-Williams MG. Programming Isotype-Specific Plasma Cell Function. *Trends Immunol.* 2019 Apr;40(4):345-357.

(2) The authors may have different views of Tfh differentiation process. Generally, PD1–CXCR5+ cells are committed Tfh cells, not pre-Tfh cells, PD1+CXCR5+ cells are GC Tfh cells, and the interconversion between these two subsets can be quite dynamic. The data actually showed that both subsets can produce IFN- γ .

Response:

We agree with the reviewer that there are divergent views on the differentiation process of Tfh cells and how these cells are characterized phenotypically. As far as we can find in published literature, there are several studies describing pre-Tfh cells as being residing in the T-B border where they meet B cells and form stable T-B conjugates that migrate into the germinal center [1-6]. These pre-Tfh cells are characterized as being CXCR5^{medium} PD-1^{medium} on flow cytometry analyses [1,2,3]. Similarly, there are other studies using CXCR5^{high} PD-1^{high} as the phenotypic marker for Tfh cells within the CD4⁺ T cell population [7-12]. Therefore, in our study, we marked pre-Tfh as CXCR5^{medium} PD-1^{medium}, and Tfh as CXCR5^{high} PD-1^{high}.

References cited:

1. Lee SK, Silva DG, Martin JL, Pratama A, Hu X, Chang PP, Walters G, Vinuesa CG. Interferon- γ excess leads to pathogenic accumulation of follicular helper T cells and germinal centers. *Immunity.* 2012 Nov 16;37(5):880-92.

2. Fujikura D, Ikesue M, Endo T, Chiba S, Higashi H, Uede T. Death receptor 6 contributes to autoimmunity in lupus-prone mice. *Nat Commun.* 2017 Jan 3; 8:13957.
3. Brian J. Laidlaw, Yisi Lu, Robert A. Amezcua, Jason S. Weinstein, Jason A. Vander Heiden, Namita T. Gupta, Steven H. Kleinstein, Susan M. Kaech, and Joe Craft. Interleukin-10 from CD4⁺ follicular regulatory T cells promotes the germinal center response. *Sci Immunol.* 2017 Oct 20; 2(16): eaan4767.
4. Nus M, Sage AP, Lu Y, Masters L, Lam BYH, Newland S, Weller S, Tsiantoulas D, Raffort J, Marcus D, Finigan A, Kitt L, Figg N, Schirmbeck R, Kneilling M, Yeo GSH, Binder CJ, de la Pompa JL, Mallat Z. Marginal zone B cells control the response of follicular helper T cells to a high-cholesterol diet. *Nat Med.* 2017 May;23(5):601-610.
5. Wing JB, Kitagawa Y, Locci M, Hume H, Tay C, Morita T, Kidani Y, Matsuda K, Inoue T, Kurosaki T, Crotty S, Coban C, Ohkura N, Sakaguchi S. A distinct subpopulation of CD25⁺ T-follicular regulatory cells localizes in the germinal centers. *Proc Natl Acad Sci U S A.* 2017 Aug 1; 114(31).
6. Woong-Kyung Suh. Life of T Follicular Helper Cells. *Mol Cells.* 2015 Mar 31; 38(3): 195–201.
7. Papa I, Saliba D, Ponzoni M, Bustamante S, Canete PF, Gonzalez-Figueroa P, McNamara HA, Valvo S, Grimbaldeston M, Sweet RA, Vohra H, Cockburn IA, Meyer-Hermann M, Dustin ML, Doglioni C, Vinuesa CG. TFH-derived dopamine accelerates productive synapses in germinal centres. *Nature.* 2017 Jul 20; 547(7663): 318–323.
8. Liu X, Chen X, Zhong B, Wang A, Wang X, Chu F, Nurieva RI, Yan X, Chen P, van der Flier LG, Nakatsukasa H, Neelapu SS, Chen W, Clevers H, Tian Q, Qi H, Wei L, Dong C. Transcription factor achaete-scute homologue 2 initiates follicular T-helper-cell development. *Nature.* 2014 Mar 27;507(7493):513-8.
9. Pratama A, Srivastava M, Williams NJ, Papa I, Lee SK, Dinh XT, Hutloff A, Jordan MA, Zhao JL, Casellas R, Athanasopoulos V, Vinuesa CG. MicroRNA-146a regulates ICOS–ICOSL signalling to limit accumulation of T follicular helper cells and germinal centres. *Nat Commun.* 2015 Mar 6; 6: 6436.
10. Yu D, Rao S, Tsai LM, Lee SK, He Y, Sutcliffe EL, Srivastava M, Linterman M, Zheng L, Simpson N, Ellyard JI, Parish IA, Ma CS, Li QJ, Parish CR, Mackay CR, Vinuesa CG. The transcriptional repressor Bcl-6 directs T follicular helper cell lineage commitment. *Immunity.* 2009 Sep 18;31(3):457-68.
11. Kitano M, Moriyama S, Ando Y, Hikida M, Mori Y, Kurosaki T, Okada T. Bcl6 protein expression shapes pre-germinal center B cell dynamics and follicular helper T cell heterogeneity. *Immunity.* 2011 Jun 24;34(6):961-72.
12. Weinstein JS, Laidlaw BJ, Lu Y, Wang JK, Schulz VP, Li N, Herman EI, Kaech SM, Gallagher PG, Craft J. STAT4 and T-bet control follicular helper T cell development in viral infections. *J Exp Med.* 2018 Jan 2;215(1):337-355.

(3) The authors focused on a specific role of IgG2c class-switch. Based on the data using CD4Cre+Tbet fl/fl and Ifngr1 KO mice, both IgG2b and 2c were affected. The germline deletion of Tbet had a more specific effect on IgG2c. These need to be reconciled.

Response:

The reviewer has rightly picked out an issue that puzzled us as well. Before obtaining direct experimental evidence, we reasoned that class-switch recombination (CSR) for IgG2c, and IgG2b may be mediated by different cytokines, and that the germline deletion of T-bet affected some, but not other cytokines. This view is more clearly presented in a recent review article by McHeyzer-Williams and colleagues in that CSR of IgG2c (or IgG2a) is facilitated by IFN- γ , whereas IgG2b by TGF- β [1]. Upon more careful literature review, we found several previous papers reporting that both IFN- γ and TGF- β signaling could modulate IgG2b class switching [2,3,4,5].

To more directly investigate the molecular events in relation to this question, we performed new experiments in a cleaner model of mixed bone marrow chimera by giving *Rag1*^{-/-} mice an equal

number cells from of *WT*: *Tbx21*^{-/-}. These chimeras will allow dissection of whether B cells or Th1-like Tfh cells function as mediators of IgG2b production. As expected, there is no Th1-like Tfh cells in *Tbx21*^{-/-} (*CD45.2*⁺) mice (**Reb. Figure 4a**). Functionally, there is no difference in IgG2b production between *WT* B cells and *Tbx21*^{-/-} B cells (**Reb. Figure 4b**). Our new data and previous studies support the idea that B cell-intrinsic T-bet is not responsible for IgG2b class switching [6,7]. Therefore, the differential effects on IgG2b and IgG2c observed previously in mice with germline deletion of T-bet may be resulted from factors produced by in non-T and non-B *Tbx21*^{-/-} cells, such as *Tbx21*-expressing NK cells, *Tbx21*-expressing dendritic cells and *Tbx21*-expressing ILC1. A definitive answer to this question will require further research.

Reb. Figure 4. B cell- intrinsic T-bet expression is not responsible for IgG2b class switching. *Rag1*^{-/-} mice receiving equal numbers of cells of *Cd45.1*: *Tbx21*^{-/-} to form mixed bone marrow chimera. The chimeric mice were infected by ZIKV with anti-IFNAR1 blocking antibody administered at one day prior to infection. Spleen cells were collected at 14 dpi for the assessment of Th1-like Tfh responses and IgG2b class switching (n=3). **(a)** Representative flow cytometry plots of IFN- γ expressing cells in *WT* and *Tbx21*^{-/-} Tfh cells (Left panel); bar graphs summarized the percentages (Right panel). **(b)** Representative flow cytometry plots of the reconstitution of *CD45.2*⁺ (*Tbx21*^{-/-}) and *CD45.2*⁻ (*Cd45.1*) in *B220*⁺ cells; Representative flow cytometry plots of IgG2b staining of *IgD*^{low} B cells (Left panel); bar graphs summarized the percentages (Right panel). The summary data were presented as mean \pm SEM. Statistical differences were determined by Student's t test and p values were indicated by * (p<0.05), or ** (p<0.01), or ***(p<0.001).

References cited:

- Higgins BW, McHeyzer-Williams LJ, McHeyzer-Williams MG. Programming Isotype-Specific Plasma Cell Function. *Trends Immunol.* 2019 Apr;40(4):345-357.
- Mohr E, Cunningham AF, Toellner KM, Bobat S, Coughlan RE, Bird RA, MacLennan IC, Serre K. IFN- γ produced by CD8 T cells induces T-bet-dependent and -independent class switching in B cells in responses to alum-precipitated protein vaccine. *Proc Natl Acad Sci U S A.* 2010 Oct 5;107(40):17292-7.
- Domeier PP, Chodiseti SB, Soni C, Schell SL, Elias MJ, Wong EB, Cooper TK, Kitamura D, Rahman ZS. IFN- γ receptor and STAT1 signaling in B cells are central to spontaneous germinal center formation and autoimmunity. *J Exp Med.* 2016 May 2;213(5):715-32.
- Coffman RL, Lebman DA, Shrader B. Transforming growth factor beta specifically enhances IgA production by lipopolysaccharide-stimulated murine B lymphocytes. *J Exp Med.* 1989 Sep 1;170(3):1039-44.
- McIntyre TM, Klinman DR, Rothman P, Lugo M, Dasch JR, Mond JJ, Snapper CM. Transforming growth factor beta 1 selectivity stimulates immunoglobulin G2b secretion by lipopolysaccharide-activated murine B cells. *J Exp Med.* 1993 Apr 1;177(4):1031-7.

6. Rubtsova K, Rubtsov AV, van Dyk LF, Kappler JW, Marrack P. T-box transcription factor T-bet, a key player in a unique type of B-cell activation essential for effective viral clearance. *Proc Natl Acad Sci U S A*. 2013 Aug 20;110(34):E3216-24.
7. Barnett BE, Staupe RP, Odorizzi PM, Palko O, Tomov VT, Mahan AE, Gunn B, Chen D, Paley MA, Alter G, Reiner SL, Lauer GM, Teijaro JR, Wherry EJ. Cutting Edge: B Cell-Intrinsic T-bet Expression Is Required To Control Chronic Viral Infection. *J Immunol*. 2016 Aug 15;197(4):1017-22.

Comments from Reviewer #2

The authors report that Th1-like Tfh cells, which co-express transcription factors T-bet and Bcl6 and produce IFN γ , are required to induce protective IgG2 antibody responses against Zika virus (ZIKV) infection. Overall, this manuscript reports an important study of the T cell requirements for inducing protective antibody responses against a clinically relevant virus. (Q1) However, the study could be strengthened by the use of cleaner models to pinpoint the role of Th1-like Tfh cells (Q2) and where IFN γ signaling is required. (Q3) Also, this work represents only a minor advance from previously reported work on the role of Tbet in Tfh function (ref 15, Weinstein et al., JEM, 2018) and it is not clear how distinct these “Th1-like Tfh” cells are from “ex-Tfh1” cells previously reported (Ref 14, Fang et al., JEM, 2018).

Our response:

We thank the reviewer for acknowledging our study showed something important. We believe the reviewer also asked us to address 3 issues, to which we respond as the following:

Response to Q1 (However, the study could be strengthened by the use of cleaner models to pinpoint the role of Th1-like Tfh cells):

As the reviewer has suggested, we first re-analyze our previous data and found no decrease in the numbers of non-Th1-like Tfh cell and non-Th1-like pre-Tfh cells in *Tbx21*^{-/-} and *Tbx21*^{f/f}*Cd4-Cre* mice (**Reb. Figure 5a, 5b**). These indicate the loss of Th1-like Tfh (and Th1-like pre-Tfh) cells is solely responsible for the decreased cell counts in *Tbx21*^{-/-} and *Tbx21*^{f/f}*Cd4-Cre* mice. To more definitively address this issue, we took the reviewer’s suggestion and performed new experiments in cleaner models of mixed bone marrow chimera by giving *Tcrb*^{-/-} two different mixture of cells: equal numbers of cells from *WT: Bcl6*^{fl/fl}*Cd4-Cre*, or *Tbx21*^{-/-: Bcl6}^{fl/fl}*Cd4-Cre*. As expected, in *Tbx21*^{-/-: Bcl6}^{fl/fl}*Cd4-Cre* mixed bone marrow chimeric mice, whose Th1-like Tfh cells are deficient (**Reb. Figure 5c, 5d; Reb. Figure 1a, 1b**), IgG2c production is impaired, whereas IgG1 production is increased (**Reb. Figure 5e, 5f; Reb. Figure 1c, 1d**). These new data, together with our previous data, more definitively showed that Th1-like Tfh cells (and Th1-like pre-Tfh cells), but not non-Th1-like Tfh cells, are responsible for IgG2c class switching. Reb. Figure 5c has been included in the revised paper as supplementary Figure 7c; Reb. Figure 5d, 5e, 5f have been included as revised Figure 5e, 5f, 5g and supplementary Figure 7d.

Reb. Figure 5. Th1-like Tfh cells are responsible for IgG2c class switching. (a) Bar graphs summarized the numbers of non-Th1-like Tfh cells in *WT* and *Tbx21*^{-/-} mice. (b) Bar graphs summarized the numbers of non-Th1-like Tfh cells in *Tbx21*^{fl/fl} and *Tbx21*^{fl/fl} *CD4-Cre* mice. (c-f) Mixed bone marrow chimeras (*WT*: *Bcl6*^{fl/fl} *Cd4-Cre*; and *Tbx21*^{-/-}: *Bcl6*^{fl/fl} *Cd4-Cre*) were administered with anti-IFNAR1 antibody one day prior to ZIKV infection. Splenocytes and sera were collected at 14 dpi for measurement of Tfh cell and antibody responses, respectively (n=5 for *WT*: *Bcl6*^{fl/fl} *Cd4-Cre* and n=4 for *Tbx21*^{-/-}: *Bcl6*^{fl/fl} *Cd4-Cre*). (c) Representative flow cytometry plots of CXCR5^{high}PD-1^{high} Tfh cells and CXCR5^{medium}PD-1^{medium} pre-Tfh cells (left panel); the percentages were summarized by bar graphs (right panel). (d) Representative flow cytometry plots of IFN-γ staining in CXCR5^{high}PD-1^{high} Tfh cells (left panel); the percentages and numbers of IFN-γ⁺ Tfh cells were summarized by bar graphs (right panel). (e) Representative flow cytometry plots of IgG1 and IgG2c intracellular staining in B220^{low}CD138⁺IgD^{low} cells (left panel); bar graphs summarized the numbers of IgG1⁺ and IgG2c⁺ cells (right panel). (f) The concentrations of ZIKV envelope specific IgG1 and IgG2c antibodies in chimeras were measured by ELISA. The summary data were presented as mean ± SEM. Statistical differences were determined by Student's t test and p values were indicated by * (p<0.05), or ** (p<0.01), or ***(p<0.001).

Response to Q2 (whether IFN γ signaling is required)

We first reanalyze our previous data and found that Th1-like Tfh cells have normal differentiation in *Ifngr1*^{-/-} (**Reb. Figure 6a**), indicating that IFN- γ signaling has no effect on Th1-like Tfh cell differentiation.

To elucidate mechanistic details, we performed new experiments using a cleaner model mixed bone marrow chimera by giving *Rag1*^{-/-} mice equal numbers of cells from *ly5.1* and *Ifngr1*^{-/-}. The resultant chimeric mice were then infected by Zika virus. In these chimeras, IFN- γ receptor-deficient Tfh cells developed normally, with slight reduction of pre-Tfh (**Reb. Figure 6b**; **Reb. Figure 2a**). Consistently, Th1-like Tfh cells have not been affected in *Ifngr1*^{-/-} (*CD45.2*), and their numbers are similar as that in *WT* (*CD45.1*) (**Reb. Figure 6c**; **Reb. Figure 2b**), but IgG2c production is decreased (**Reb. Figure 6d**; **Reb. Figure 2c**). These data support the conclusion that IFN- γ signaling is required for B cell to mediate antibody class switching. **Reb. Figure 6a, 6b** have

been included in the revised manuscript as supplementary Figure 8a, 8c; Reb. Figure 6c, 6d have been included as revised Figure 5e, 5f.

Reb. Figure 6. IFN- γ signaling is required for B cell to mediate antibody class switching. (a) Representative flow cytometry plots of IFN- γ production in Tfh cells from *WT* and *Ifngr1*^{-/-} mice (Left panel); bar graphs summarized the percentages (Right panel). (b-d) *Rag1*^{-/-} mice receiving equal numbers of (*Cd45.1: Ifngr1*^{-/-}) mixed bone marrow cells were infected by ZIKV with anti-IFNAR1 blocking antibody at one day prior to infection. Spleens were collected at 14 dpi for assessment of Th1-like Tfh responses and antibody class switching (n=5). (b) Representative flow cytometry plots of the reconstitution of CD45.2⁺ (*Ifngr1*^{-/-}) and CD45.2⁻ (*Cd45.1*) in CD4⁺ cells; Representative flow cytometry plots of Tfh cells and pre-Tfh cells of CD4⁺ CD44^{high} CD62L^{low} cells (Left panel); bar graphs summarized the percentages (Right panel). (c) Representative flow cytometry plots of IFN- γ production in Tfh cells from CD45.2⁺ (*Ifngr1*^{-/-}) and CD45.2⁻ (*Cd45.1*) mice (Left panel); bar graphs summarized the percentages (Right panel). (d) Representative flow cytometry plots of IgG2c production in B220^{low} CD138⁺ IgD⁻ cells from CD45.2⁺ (*Ifngr1*^{-/-}) and CD45.2⁻ (*Cd45.1*) (Left panel); bar graphs summarized the percentages (Right panel). The summary data were presented as mean \pm SEM. Statistical differences were determined by Student's t test and p values were indicated by * ($p < 0.05$).

Response to Q3 (This work represents only a minor advance from previously reported work on the role of Tbet in Tfh function (ref 15, Weinstein et al., JEM, 2018) and it is not clear how distinct these “Th1-like Tfh” cells are from “ex-Tfh1” cells previously reported (Ref 14, Fang et al, JEM, 2018))

We want to emphasize that our paper is different from previous publications and that our “Th1-like Tfh” cells are from “ex-Tfh1” cells for the following reasons:

- (1) Compared with ref 15 (Weinstein et al., JEM, 2018) which demonstrates the role of STAT4 and Tbet in Tfh development during LCMV infection, our study used acute Zika virus infection model and demonstrated that Th1-like Tfh cells are functionally essential for both the induction of long-term ZIKV-specific protective antibody response, and IgG2c antibody class switching.
- (2) As to the difference between our Th1-like Tfh and previously reported ex-Tfh1 (ref.14, Fang et al, JEM, 2018) there are abundant differences.

Th1-like Tfh cells in our study are phenotypically and transcriptomically distinct from ex-Tfh1 reported in ref.14. There are 33.8% of Th1-like Tfh cells express T-bet, whereas nearly no expression of T-bet in ex-Tfh1 cells. All of Th1-like Tfh cells express IFN- γ and CXCR3, whereas only 24.2% of ex-Tfh1 cells produce IFN- γ and about 43% of them express CXCR3. And, only 11% of Th1-like Tfh cells express NKG2D, compared to 87% of ex-Tfh1 cells doing (**Reb. Figure 7a**). We also compared the RNA-seq data from tableS2 in ref14 (Fang et al., JEM, 2018) with that of ours. In Th1-like Tfh cells, there are 104 unique up-regulated genes and 289 unique down-regulated genes, whereas ex-Tfh1 cells have 177 unique up-regulated genes and 32 unique down-regulated genes, the numbers do not match. Unsurprisingly, however, Th1-like Tfh cells and ex-Tfh1 cells share only 19 common up-regulated genes, and only 1 common down-regulated gene compared to conventional tfh cells. (**Reb. Figure 7b**). To acquire more information, we downloaded the raw RNAseq data of ex-tfh1, after normalization with our RNAseq data, we found 831 up-regulated genes (such as *crtam* and *ly6i*) and 846 down-regulated genes (such as *il2ra* and *klrc1*) in Th1-like Tfh cells compared to ex-Tfh1 cells. The heatmap shows that ex-Tfh1 cells have lost some characteristics of Tfh cells, for instance, Tfh signature genes *Bcl6*, *Cxcr5*, *Il21* are down regulated (**Reb. Figure 7c**). In summary, Th1-like Tfh cells are not the same as ex-Tfh1 cells. Part of Reb. Figure 7a (NKG2D staining in Th1-like Tfh cells, far right panel, upper plot) has been included in the revised manuscript as supplementary Figure 4e.

Reb. Figure 7. Th1-like Tfh cells are distinct from ex-Tfh1 cells. (a) Representative flow cytometry plots of T-bet, IFN- γ , CXCR3 and NKG2D expression in Th1-like Tfh cells and ex-Tfh1 cells. (b) The up-regulated and down-regulated genes in Th1-like Tfh cells and ex-Tfh1 cells ($p < 0.05$, $F_c > 2$). (c) Heatmap of Tfh and Th1 related key genes in Th1-like Tfh cells and ex-Tfh1 cells.

Reb. Table 1 Summary of difference between Th1-like Tfh cells and ex-Tfh1 cells.

	Th1-like Tfh	ex-Tfh1
T-bet level	High	Low
IFN- γ ⁺ cells	100%	24%

CXCR3 ⁺ cells		95%~	43%
NKG2D ⁺ cells		11%	87%
Based on ref14 TableS2	Unique up-regulated genes	104	177
	Common up-regulated genes	19	
	Unique down-regulated genes	289	32
	Common down-regulated genes	1	
Based on ref14 and our original data	up-regulated genes (Th1-like Tfh vs. ex-tfh1)	831	
	down-regulated genes (Th1-like Tfh vs. ex-Tfh1)	846	

References cited:

Ref14. Fang D, Cui K, Mao K, Hu G, Li R, Zheng M, Riteau N, Reiner SL, Sher A, Zhao K, Zhu J. Transient T-bet expression functionally specifies a distinct T follicular helper subset. *J Exp Med.* 2018 Nov 5;215(11):2705-2714.

Ref15. Weinstein JS, Laidlaw BJ, Lu Y, Wang JK, Schulz VP, Li N, Herman EI, Kaech SM, Gallagher PG, Craft J. STAT4 and T-bet control follicular helper T cell development in viral infections. *J Exp Med.* 2018 Jan 2;215(1):337-355.

Q4: They perform their experiments in immunocompetent mice, which are normally resistant to ZIKV infection due to type I IFN responses, and therefore rely on a model of anti-IFNAR1 blockade prior to ZIKV infection. The authors nicely argue that anti-IFNAR1 is not responsible for the Th1-like Tfh cells they observe in ZIKV infection by performing parallel experiments with LCMV infection with and without anti-IFNAR1 pre-treatment. They show that LCMV infection, which can occur in mice with intact type I IFN responses, induces similar degrees of total Tfh cells and IFN γ + Tfh cells regardless of anti-IFNAR1 treatment.

Response:

We thank the reviewer for pinpoint a strength of our experimental system.

Q5: The experiments depleting Th1-like Tfh cells using total or T cell-specific Tbx21 deficiency are the main weakness of this paper. While the authors intend to use these mice to specifically deplete Th1-like Tfh cells, they find that Tbx21^{-/-} mice have decreased numbers of total pre-Tfh cells (Fig. 5a) and Tbx21f/fCD4-Cre mice have decreased numbers of both pre-Tfh cells and Tfh cells (Fig. S7a). Therefore, it is unclear whether the decreased IgG2c antibody titers are due to specific loss of Th1-like Tfh cells or just overall reduction of Tfh cells. The decrease in pre-Tfh and Tfh cell counts is potentially not an issue if the loss of Th1-like Tfh cells is solely responsible for the decreased counts – the authors could see if non-Th1-like Tfh cell counts remain the same in Tbx21-deficient mice. However, if both Th1-like and non-Th1-like Tfh cell counts are reduced, the authors would need to use a cleaner experimental model (e.g. Bcl6f/fCD4-Cre:Tbx21^{-/-} mixed bone marrow chimera mice). The authors could also demonstrate that IFN γ is the operative signal from Th1-like Tfh cells that induces IgG2 switching with the use of Bcl6f/fCD4-Cre:Ifng^{-/-} mixed bone marrow chimera mice.

Response:

We agree with the reviewer that these above underlined issues (added by the authors) are indeed important for understanding the specific molecular mechanisms of our reported observations. As the reviewer has suggested, we re-analyze our data and found no decrease in non-Th1-like Tfh cell and non-Th1-like pre-Tfh cell counts from *Tbx21*^{-/-} and *Tbx21*^{f/f}*Cd4-Cre* mice (**Reb. Figure 5a, 5b**), suggesting the loss of Th1-like Tfh (and Th1-like pre-tfh) cells is solely responsible for the decreased counts in *Tbx21*^{-/-} and *Tbx21*^{f/f}*Cd4-Cre* mice.

To further study this issue from the molecular mechanism perspective, we took this reviewer's insightful suggestion and performed new experiments in cleaner models of mixed bone marrow chimera by giving *Tcrb*^{-/-} mice different mixture of cells: equal numbers of WT: *Bcl6*^{fl/fl}*Cd4-Cre*, or *Tbx21*^{-/-}: *Bcl6*^{fl/fl}*Cd4-Cre* to form chimeric mice in which we performed Zika virus infection experiments. Detailed description of the results of these experiments can be found above in “**Response to Reviewer #2 (Q1)**” and **Reb. Figure 5**.

To investigate whether IFN- γ is the operative signal from Th1-like Tfh cells, or other cells, that induces IgG2 switching. We performed new experiments as this reviewer has suggested in cleaner models of mixed bone marrow chimera by giving *Tcrb*^{-/-} mice different mixture of cells: equal numbers of WT: *Bcl6*^{fl/fl}*Cd4-Cre*, or *Ifng*^{-/-}: *Bcl6*^{fl/fl}*Cd4-Cre* to form chimeric mice in which we performed Zika virus infection experiments. Detailed description of the results of these experiments can be found above in “**Response to Editorial comments**” and **Reb. Figure 3**.

Q6: Additionally, Figure 6 argues that IFN- γ signaling is required for class switching to IgG2, but does not pinpoint which cell type requires such signaling due to the use of total knockout *Ifngr1*^{-/-} mice. The authors do demonstrate unimpaired pre-Tfh and Tfh cell induction in *Ifngr1*^{-/-} mice, but do not demonstrate their cytokine profile – does *Ifngr1* deficiency affect the ability of Tfh cells to produce IFN γ ? Or is *Ifngr1* deficiency in B cells solely responsible for decreased IgG2 production? The authors could add clarity to this part of the manuscript by limiting *Ifngr1* deficiency to specific cell types.

Response:

We thank Reviewer #2 for the insightful and critical comment. These questions have been asked and addressed earlier. To recap, we analyze our data and found that Th1-like Tfh cells have normal differentiation in *Ifngr1*^{-/-} (**Reb. Figure 6a**), indicating IFN- γ signaling has no effect in Th1-like Tfh cells differentiation.

To further address this question, we used a cleaner model by giving *Rag1*^{-/-} equal numbers of *Iy5.1* and *Ifngr1*^{-/-} mixed bone marrow chimera to perform Zika virus infection experiment. Results of which can be found above in “**Response to Reviewer #2 (Q2)**” and **Reb. Figure 6**.

Minor comments:

1. The manuscript contains a number of typos throughout.

Response:

We have carefully rechecked our revised manuscript and corrected the typos.

2. The initial characterization of Tfh cell response and cytokine profile is performed in BALB/c mice (Fig. 1-2). However, later studies are done in Bcl6f/fCD4-Cre mice on C57BL/6 background. Do the authors have parallel characterizations in both strains? Obviously these two strains have very different T cell differentiation proclivity.

Response:

We thank Reviewer #2 for pointing out this important issue. In fact, during preliminary studies, we have done the parallel characterization in both C57BL/6 and BALB/c in T cell differentiation. Because it is easier to establish an experiment model with BALB/c mice that are easier to handle, we opted to use it during the model set-up stage. For mechanistic studies, most KO mice are in C57BL/6

Basically, the two mouse strains showed similar but not identical results. Specifically, there is a stronger Tfh response but a similar pre-Tfh response in BALB/c compared with C57BL/6 mice (**Reb. Figure 8a**). As for responses from other T helper subsets, their Th1 responses are similar (**Reb. Figure 8b**), but their IFN- γ production are slightly different. These minor differences between the mouse strains do not change the conclusions derived from our experimental data.

Reb. Figure 8. Comparison of T cell responses between C57BL/6 and BALB/c mice in ZIKA infection. C57BL/6 and BALB/c mice that were pre-treated with anti-IFNAR1 antibody were administered with PBS or ZIKV. Splenocytes were collected at 7 dpi for staining. (n=3 for each group). Representative flow cytometry plots of (a) Tfh and pre-Tfh cells, (b) IFN- γ ⁺ cells (Th1), IL-4⁺ cells (Th2) and IL-17A⁺ (Th17) cells in CD4⁺ T cells, with bar graphs summarized the percentages of each cell type. The summary data were presented as mean \pm SEM. Statistical differences were determined by Student's t test and p values were indicated by * (p<0.05), or ** (p<0.01), or *** (p<0.001).

3. Statistics are missing in most of the subfigures within Fig. 1.

Response:

We have added statistics to Fig.1 in the revised manuscript.

4. In Figure 2b, please indicate on which day Tfh cells were assessed for IFN γ production.

Response:

We have now added the time-points in our revised figure legend of Fig 2b.

5. Line 213: Gata3 is not a conventional Tfh-associated gene.

Response:

We thank the reviewer for pointing this out. We have deleted *gata3* in our revised manuscript.

Reviewer #3

This manuscript by Liang and colleagues report a role for Th1-like Tfh in shaping the humoral response to Zika virus (ZIKV) infection in a mouse model. The authors used BALB/c mice with IFNAR inhibition to enable productive ZIKV infection as evidenced by detectable viremia to identify how this subset of T cells is important in inducing robust IgG response, especially in the generation of IgG2 antibodies. The authors suggest that their findings provide insights into the targeting of Th1-like Tfh cells for manipulation to induce robust humoral response to vaccination.

The findings are indeed interesting and the authors appear to have done extensive amount of work to pinpoint an important role for this subset of Tfh cells. I appreciated the attention to detail in ruling out a possible effect of IFNAR blocking antibodies in shaping the humoral response. Likewise, the use of relevant knockout mouse models supported the line of thinking clearly. I have a few comments for the authors to consider.

Response:

We thank Reviewer #3 for the encouraging comment.

1. The authors hint at an underlying factor that could account for the difference in humoral response to ZIKV infection in IFNAR blocked vs control BALB/c mice. Viremic infection in the former likely have resulted in more antigens being expressed compared to the latter. Greater antigen load could hence have resulted in more antigen presentation by Tfh cells or activation of these cells in the lymph nodes, or both. Alternatively, the immune response could have been dependent on replicating ZIKV, where the pro-inflammatory factors expressed in infected cells could have stimulated antigen presenting cells and/or T cells in ways that cannot be reproduced even if an equivalent level of inactivated antigens were inoculated in these animals. The inclusion of an inactivated ZIKV control in the experiments could have clarified these questions. It would also inform on whether, besides the degree of cellular immunity, qualitatively different humoral response could be elicited by replicating compared to non-replicating vaccines due to the differences in Th1-like Tfh stimulation. This possibility could have important implications on how

the pipeline of candidate Zika vaccines should be prioritised for clinical development. Perhaps the authors would consider adding a discussion on this issue.

Response:

We think the reviewer’s major concern here is how antigen dose will affect the observed immune response. The following are pertinent results. We have also elaborated our thought on this point in the discussion section of our revised manuscript.

In fact, we have done an experiment with inactivated Zika virus as control. In the presence of anti-IFNAR1 antibody pre-treatment, inactivated Zika virus induced relatively fewer Tfh and Th1-like Tfh cells, or other helper T cell subsets, and much less antibody response compared with the same experiment done with live Zika virus (**Reb. Figure 9**).

Reb. Figure 9. Minimal T cell response and antibody response in mice administered with inactivated ZIKA. C57BL/6 mice were administered with PBS, inactivated ZIKV or live ZIKV with anti-IFNAR1 antibody pre-treatment. Splenocytes were collected at 7 dpi for Tfh and Th1-like Tfh staining (n=3 for PBS group, and n=4 for inactivated ZIKA and live ZIKV group); and in a parallel experiment, splenocytes were collected at 14 dpi for other helper T cell subsets cytokines and antibody response staining. Representative flow cytometry plots of (a) Tfh and Pre-Tfh cells, (b) IFN-γ production in Tfh cells, (c) IFN-γ+ cells (Th1), IL-4+ cells (Th2) and IL-17A+ (Th17) cells in CD4+ T cells, (d) IgM, IgG1, IgG2b and IgG2c response in IgD^{low} B cells, with bar graphs summarized of the percentages. The summary data were presented as mean ± SEM. Statistical

differences were determined by Student's t test and p values were indicated by * ($p < 0.05$), or ** ($p < 0.01$), or *** ($p < 0.001$).

2. Line 441 indicated that 500-1000 pfu/ml virus was incubated with an equivalent volume of serially diluted serum before inoculation onto a monolayer of Vero cells in a 24-well plate. Unless the volume was very small, which could introduce inaccuracies, the number of pfu appears to be very large for a 24-well plate assay. Is this description correct? Please clarify.

Response:

We thank Reviewer #3 for mentioning this issue. For plaque reduction neutralization test (PRNT), Zika virus (50-100 PFU/100ul/well) was mixed with serial diluted inactivated serum at a volume ratio of 1:1. The actual volume is indeed small, each well is added with add 200 μ l of Virus+Serum mixture that contains 0-100 PFU of the virus. We have also added detailed description of the assay in the method section of revised manuscript.

3. Figure 1a suggests that there is some background ZIKV neutralization activity in the uninfected animal controls (a 50% neutralization titer of nearly 2 logs at 14 days). This is rather unusual. Can the authors explain what is going on?

Response:

We thank Reviewer #3 for pointing out this issue. We observed that the PRNT50 values in uninfected mice at day14 is a little higher in one of the three mice, whereas results in other two mice are similar to normal control levels (**Original Figure 1b, which is now Reb. Figure 10a**). This apparent outlier has swayed the data off a bit.

To further verify the observed higher background neutralization in the uninfected mice is indeed spurious, we performed new experiments using both normal uninfected C57BL/6 and BALB/c mice, which also showed generally low PRNT50 levels, except for that at high concentration (**Reb. Figure 10b, 10c**). So, it is likely that the higher background neutralization activity at 14 dpi in one of the mice was due to individual variation among mice, but not a systemic problem with the PRNT assay.

Reb. Figure 10. Background neutralization of PBS treated mice to Zika virus. (a) BALB/c mice were administered with PBS control (Uninfected), ZIKV only (ZIKV) or ZIKV+ anti-IFNAR1 receptor antibody (ZIKV + α -IFNAR1), the kinetics of sera neutralizing antibody responses were examined on days 7, 14 and 28 post infection (dpi) (n=3 for each group). (b-c) BALB/c and C57BL/6 mice were administered with PBS and sera

were collected at 14 days, Sera is inactivated and neutralization activity to Zika virus was tested on Vero monolayer as described. **(b)** Neutralization activity of serum from uninfected C57BL/6 and BALB/c to ZIKV; **(c)** PRNT50 of uninfected C57BL/6 and BALB/c mice serum to ZIKV;

4. Line 155. Figure 2b should be labelled here as Figure 3b.

Response:

We check the original submission for Figure 2b at line 155, which looks like a correct label. In contrast, Figure 3b shows the Scatterplot analysis of the expression of Tfh (left) and Th1 (right) -related genes in Th1-like Tfh cells compared with conventional Tfh and Th1 cells. These figures are shown below for comparison.

line 155:

154 control (10%), those inoculated with inactivated ZIKV (19.7%), or ZIKV only
 155 (35.8%) **(Fig. 2b)** **(Representative FACS plot on the Left, and summary**
 156 **data on the Right)**. Since Tfh cell surface marker CXCR5 is susceptible to

Figure 2b:

Figure 3b:

5. There are scattered typographical/grammatical errors in the manuscript.

Response:

We have rechecked our manuscript carefully and corrected all the typos in the revised version.

Reviewers' Comments:

Reviewer #1:

Remarks to the Author:

I have reviewed the point-by-point responses to my comments and those by the other two reviewers. Although conceptually the recognition of IFN γ -producing Tfh cells is not entirely novel, considering that the response is observed in clinically relevant zika virus, and the portion of Th1-like Tfh cells is substantially higher than other model viruses, the study has its unique value in informing the field.

The authors also put good-faith efforts to address all reviewers' concerns, especially the generation of various mixed BM chimeras helped clarify several issues.

In addition, comparative analysis of transcriptomes between Th1-like Tfh cells with ex-Tfh1 cells reveals unique molecular features. This aspect does add novelty to the authors' findings, and I recommend inclusion of Figure 7 in the rebuttal letter as supplemental figures, and proper emphasis on this finding.

Overall, I am supportive of publication of this work in N Communs.

Reviewer #2:

Remarks to the Author:

We commend the authors for their revised data provided to the reviewers, which provides clearer mechanistic insight into the role of Th1-like Tfh cells in inducing protective antibody responses against Zika virus. We particularly appreciate their use of multiple bone marrow chimeric mice to specifically delete the Th1-like Tfh population, to specifically delete IFN γ from Tfh cells, and to isolate IFN γ signaling to specific cell types. We also appreciate the authors' computational analysis of Th1-like Tfh cells vs ex-Tfh1 cells. HOWEVER, after spending a significant amount of time, I could not find most of the "reviewer figures" in the revised manuscript (main figs or supplement). This is unacceptable. The review process is for a revised public not private manuscript. If I am correct only the data from Fig S4d-e has been added based on our previous comments. As such, I do not find the revised manuscript significantly improved from the first round of reviews.

In addition to the data provided to the reviewers being added to the full manuscript, additional minor issues should be addressed:

Could the authors show the following pieces of data from their bone marrow chimeric mice?

1. The authors should demonstrate similar frequencies/counts of IFN γ + Th1 cells in WT:Bcl6fl/fICd4-Cre vs Tbx21-/-:Bcl6fl/fICd4-Cre mice, as well as WT:Bcl6fl/fICd4-Cre vs Ifng-/-: Bcl6fl/fICd4-Cre mice. This would be an important control to demonstrate that these bone marrow chimeric mice indeed isolate Tbet or IFN γ deficiency to the Tfh cell compartment.
2. The authors should show reconstitution frequencies of WT vs Ifngr1-/- B cells in the WT: Ifngr1-/- mice. Additionally, could they show %IgG1+ B cells? In the full Ifngr1-/- mice, the authors demonstrate decreased IgG2c switching and increased IgG1 switching. Is this phenocopied in a B cell-intrinsic manner?

Reviewer #3:

Remarks to the Author:

In this revised submission, Liang and colleagues have addressed most of the concerns raised in their original manuscript. The authors have also provided a very detailed rebuttal, supported by additional data to substantiate their responses. I appreciate their efforts to conduct additional experiments to address the reviewers' concerns.

Minor comments:

1. There are still quite a number of typographical errors that need to be corrected.
2. Lines 66-67. I believe the authors are referring to neutralizing antibodies in mice. If so, I suggest amending the sentence to "... antibody subclass following viral infections in mice." The predominant IgG subclass following viral infections in humans are virus dependent, in contrast. Eg IgG1 appears to be the predominant IgG subclass after adenovirus infection (Murphy et al, J Med Virol 2009) whereas measles induce mostly IgG3 (Toptygina et al, Clin Diag Lab Immunol 2005).
3. Line 136. I believe the authors are referring to Figures 2a and 2b and not 1a and 1b.
4. Line 432. Change ZIKA virus to ZIKV.
5. Line 456. Insert "in the" between "residing" and 'T-B border'.
6. Line 461. Change "nature" to "natural".
7. Line 473. Change "them" to "such vaccines".
8. Finally, I would suggest that the authors consider including Rebuttal Figure 9 as a supplementary figure in the manuscript. The data is interesting and goes towards supporting the narrative that ZIKV infection and replication are needed to induce the observed adaptive immune responses.

Response to reviewers' comments

Comments from Reviewer #1

I have reviewed the point-by-point responses to my comments and those by the other two reviewers. Although conceptually the recognition of IFN γ -producing Tfh cells is not entirely novel, considering that the response is observed in clinically relevant zika virus, and the portion of Th1-like Tfh cells is substantially higher than other model viruses, the study has its unique value in informing the field.

Response:

We thank the reviewer for noticing our study has value.

The authors also put good-faith efforts to address all reviewers' concerns, especially the generation of various mixed BM chimeras helped clarify several issues.

Response:

We thank the reviewer for acknowledging the effort we put into improving our paper.

In addition, comparative analysis of transcriptomes between Th1-like Tfh cells with ex-Tfh1 cells reveals unique molecular features. This aspect does add novelty to the authors' findings, and I recommend inclusion of Figure 7 in the rebuttal letter as supplemental figures, and proper emphasis on this finding.

Overall, I am supportive of publication of this work in N Communs.

Response:

We thank Reviewer #1 for the suggestion. We have included the previous Reb. Figure 7 as supplemental figure 6, except for the lower panel of the previous Reb. Figure 7a, which has been published by others. The new data have been described in some length in the revised manuscript.

Supplementary figure 6. Th1-like Tfh cells are distinct from “ex-T-bet” Tfh cells. (a) The up-regulated and down-regulated genes in Th1-like Tfh cells and “ex-T-bet” Tfh cells ($p < 0.05$, $F_c > 2$). **(b)** Heatmap of Tfh and Th1 related key genes in Th1-like Tfh cells and “ex-T-bet” Tfh cells. **(c)** Representative flow cytometry plots of T-bet, IFN- γ , CXCR3 and NKG2D expression in Th1-like Tfh cells.

Comments from Reviewer #2

We commend the authors for their revised data provided to the reviewers, which provides clearer mechanistic insight into the role of Th1-like Tfh cells in inducing protective antibody responses against Zika virus. We particularly appreciate their use of multiple bone marrow chimeric mice to specifically delete the Th1-like Tfh population, to specifically delete IFN γ from Tfh cells, and to isolate IFN γ signaling to specific cell types. We also appreciate the authors’ computational analysis of Th1-like Tfh cells vs ex-Tfh1 cells.

Response:

We thank the reviewer for these encouraging remarks.

HOWEVER, after spending a significant amount of time, I could not find most of the “reviewer figures” in the revised manuscript (main figs or supplement). This is unacceptable. The review process is for a revised public not private manuscript. If I am correct only the data from Fig S4d-e has been added based on our previous comments. As such, I do not find the revised manuscript significantly improved from the first round of reviews.

Response:

We apologize for not having demonstrated our effort more noticeably in the previous revision. We thank the reviewer for suggestion of adding the “reviewer figures” to the main text. In the revised paper, we added two new figures, **Fig.6 and Fig.8**; and **supplementary Fig. 3 (a-c), supplementary Fig. 6 (a-c), and supplementary Fig. 10a, 10c.**

In addition to the data provided to the reviewers being added to the full manuscript, additional minor issues should be addressed:

Could the authors show the following pieces of data from their bone marrow chimeric mice?

1. The authors should demonstrate similar frequencies/counts of IFN γ ⁺ Th1 cells in WT:Bcl6fl/flCd4-Cre vs *Tbx21*^{-/-}:Bcl6fl/flCd4-Cre mice, as well as WT:Bcl6fl/flCd4-Cre vs *Ifng*^{-/-}: Bcl6fl/flCd4-Cre mice. This would be an important control to demonstrate that these bone marrow chimeric mice indeed isolate Tbet or IFN γ deficiency to the Tfh cell compartment.
2. The authors should show reconstitution frequencies of WT vs *Ifngr1*^{-/-} B cells in the WT: *Ifngr1*^{-/-} mice. Additionally, could they show %IgG1⁺ B cells? In the full *Ifngr1*^{-/-} mice, the authors demonstrate decreased IgG2c switching and increased IgG1 switching. Is this phenocopied in a B cell-intrinsic manner?

Response:

We thank Reviewer #2 for pointing out these important issues.

(1) We reanalyzed our data in mixed bone marrow experiments and found no significant reduction in the percentage of IFN- γ ⁺ Th1 cells in the *Tbx21*^{-/-}: *Bcl6*^{fl/fl} *Cd4*-Cre chimeric mice (**Reb. Figure 1a**), and *Ifng*^{-/-}: *Bcl6*^{fl/fl} *Cd4*-Cre chimeric mice (**Reb. Figure 1b**) compared with WT:

Bcl6^{f/f}Cd4-Cre, indicating these bone marrow chimeric mice indeed isolate T-bet or IFN- γ deficiency to the Tfh cell compartment. These results also can be found above in “**Response to Editorial comments (2)**”. Reb. Figure 1a has been included as revised Figure 6a; Reb. Figure 1b has been included as revised Figure 8f.

(2) We also reanalyzed our data in mixed bone marrow (*CD45.1⁺ WT: Ifngr1^{-/-}*) experiments and showed there are similar reconstitution frequencies between *CD45.1⁺ WT* B cells and *Ifngr1^{-/-}* B cells (**Reb. Figure 2a**), which indicates that the reconstitution of *CD45.1⁺ WT* and *Ifngr1^{-/-}* B cells is normal. In addition, we also found the increased IgG1 switching in *Ifngr1^{-/-}* in B220^{low} CD138⁺ IgD^{low} cells (**Reb Figure 2b**), consistent with our previous data in the full *Ifngr1^{-/-}* mice. Together, these results indicate that the increased IgG1 switching is B-cell intrinsic. These results also can be found above in “**Response to Editorial comments (2)**”. Reb. Figure 2a has been included as part of revised supplementary Fig. 10c; Reb. Figure 2b has been included as revised Figure 8d.

Comments from Reviewer #3

Reviewer #3 (Remarks to the Author):

In this revised submission, Liang and colleagues have addressed most of the concerns raised in their original manuscript. The authors have also provided a very detailed rebuttal, supported by additional data to substantiate their responses. I appreciate their efforts to conduct additional experiments to address the reviewers’ concerns.

Response:

We thank the reviewer for an overall enthusiastic comment.

Minor comments:

1. There are still quite a number of typographical errors that need to be corrected.
2. Lines 66-67. I believe the authors are referring to neutralizing antibodies in mice. If so, I suggest amending the sentence to “... antibody subclass following viral infections in mice.” The predominant IgG subclass following viral infections in humans are virus dependent, in contrast. Eg IgG1 appears to be the predominant IgG subclass after adenovirus infection (Murphy et al, J Med Virol 2009) whereas measles induce mostly IgG3 (Toptygina et al, ClinDiag Lab Immunol 2005).
3. Line 136. I believe the authors are referring to Figures 2a and 2b and not 1a and 1b.
4. Line 432. Change ZIKA virus to ZIKV.
5. Line 456. Insert “in the” between “residing” and ‘T-B border’.
6. Line 461. Change “nature” to “natural”.
7. Line 473. Change “them” to “such vaccines”.

Response:

We apologize for the above mistakes and have corrected them as suggested.

8. Finally, I would suggest that the authors consider including Rebuttal Figure 9 as a supplementary figure in the manuscript. The data is interesting and goes towards supporting the narrative that ZIKV infection and replication are needed to induce the observed adaptive immune responses.

Response:

As suggested, we have included this figure as a supplementary figure in the revised manuscript, except for Rebuttal Figure 9b, which has been included in Figure 2b in the original manuscript.

Reviewers' Comments:

Reviewer #2:

Remarks to the Author:

We appreciate the authors' inclusion of their previous rebuttal analysis of Th1-like Tfh cells vs ex-Tfh1 cells in the newly revised manuscript, as well as additional analysis of the bone marrow chimeric mice. We recommend the following changes to the manuscript before publication:

1. Supp Fig 6: For panel C, which demonstrates lack of ex-Tfh1 markers on Th1-like Tfh cells, the authors should show a positive control of staining of these markers (esp. NKG2D) on ex-Tfh1 cells. The authors have previously shown such data in the bottom half of Rebuttal Fig 7a in their first rebuttal. Additionally, for the legend of this figure (line 92), they should correct "heatmap" to "heatmap."
2. The current title for Fig 5 & Supp Fig 9 (Th1-like Tfh is required for IgG2c antibody class switching) is more appropriate for Fig 6, as Fig 5 & Supp Fig 9 use models that knock out Tbet in all cells or all T cells. Only Fig 6 shows data for specific knockout of Th1-like Tfh cells, using bone marrow chimeric mice to selectively delete Tbet in Tfh cells. The titles for Fig 5 & Supp Fig 9 should therefore be revised to reflect the data shown.
3. We recommend that the titles for Fig 7 & Supp Fig 10 be revised as follows to more accurately reflect the data shown: The IFN- γ pathway is required for IgG2c antibody class-switching.
4. We recommend that the title for Fig 8 be revised as follows to more accurately reflect the data shown: IFN- γ produced by Th1-like Tfh cells is required in a B cell-intrinsic manner for IgG2c antibody class-switching.

Response to reviewers' comments

Editorial comments:

Your manuscript entitled "ZIKV infection induces robust Th1-like Tfh cell and long-term protective antibody responses in immunocompetent mice" has now been seen again by our referees, whose comments appear below. In light of their advice I am delighted to say that we are happy, in principle, to publish a suitably revised version in Nature Communications under the open access CC BY license (Creative Commons Attribution v4.0 International License).

Response:

We thank you very much for considering publishing our revised paper.

We therefore invite you to revise your paper one last time to address the remaining concerns of our reviewers. At the same time we ask that you edit your manuscript to comply with our format requirements and to maximise the accessibility and therefore the impact of your work.

Response:

We have responded to the remaining concerns of the reviewers. Additionally, we checked all the format requirements of the journal and revised our paper wherever appropriate.

Comments from Reviewer #2

We appreciate the authors' inclusion of their previous rebuttal analysis of Th1-like Tfh cells vs ex-Tfh1 cells in the newly revised manuscript, as well as additional analysis of the bone marrow chimeric mice. We recommend the following changes to the manuscript before publication:

1. Supp Fig 6: For panel C, which demonstrates lack of ex-Tfh1 markers on Th1-like Tfh cells, the authors should show a positive control of staining of these markers (esp. NKG2D) on ex-Tfh1 cells. The authors have previously shown such data in the bottom half of Rebuttal Fig 7a in their first rebuttal.

Response:

We interpret the reason for requesting positive control staining, especially for NKG2D, is a concern that our Th1-like Tfh cells were false-negative for this signature marker of ex-Tfh1, and therefore a key difference between these two type of cells may not exist. To this, we have prepared two answers, each of which can independently address the reviewer's concern.

The first one is as the reviewer has suggested by simply showing Rebuttal Fig 7a in our first rebuttal letter as part of a supplementary figure in this revision. To do this, however, will need to ask for copy right permission from JEM where the ex-Tfh1 data were published. Some of that published data were reanalyzed for making comparison with our Th1-like Tfh results, and used in Rebuttal Fig 7a. Therefore, we have made a request to JEM for permission to use their published materials in the format we intended to use in our current paper. This process is expected to take more than the two-week limit suggested for resubmission. If the editor strongly advise us to take

this route, we would like to ask for an extension for resubmission until we heard an affirmative reply from JEM.

An alternative approach, which we think can also satisfactorily address the reviewer's concern. We reason that if we can demonstrate the NKG2D antibody can stain positively on cells that are expected to express high levels of this protein, such as NK cells, and then a negative or weakly positive staining on Th1-like Tfh with the same antibody is unlikely to be a false negative result. In the current revision, we provided new data in the form of supplementary Fig.6d to show that NKG2D antibody we used can indeed stain NK cells nicely. We revised corresponding text section to reflect the new results.

In the interests of returning a revised version of our paper within two weeks, we submitted the revised paper with the second approach.

2. Additionally, for the legend of this figure (line 92), they should correct "heapmap" to "heatmap."

Response:

We have revised the mistake accordingly.

3. The current title for Fig 5 & Supp Fig 9 (Th1-like Tfh is required for IgG2c antibody class switching) is more appropriate for Fig 6, as Fig 5 & Supp Fig 9 use models that knock out Tbet in all cells or all T cells. Only Fig 6 shows data for specific knockout of Th1-like Tfh cells, using bone marrow chimeric mice to selectively delete Tbet in Tfh cells. The titles for Fig 5 & Supp Fig 9 should therefore be revised to reflect the data shown.

Response:

We agree with these suggestions and made all changes accordingly.

4. We recommend that the titles for Fig 7 & Supp Fig 10 be revised as follows to more accurately reflect the data shown: The IFN- γ pathway is required for IgG2c antibody class-switching.

Response:

We agree and made changes accordingly.

5. We recommend that the title for Fig 8 be revised as follows to more accurately reflect the data shown: IFN- γ produced by Th1-like Tfh cells is required in a B cell-intrinsic manner for IgG2c antibody class-switching.

Response:

We agree and made changes accordingly.